# FedRGL: Robust Federated Graph Learning under Label Noise

De Li [1 2]  Zhou Tan [3]  Qiyu Li [1 4]  Zeming Gan [1 4]  Tiange Xia [1]  Chunpei Li [1 4]  Xianxian LI [1 4]

## Abstract

Federated Graph Learning (FGL) is a distributed machine learning paradigm based on graph neural networks, enabling secure and collaborative modeling of local graph data among clients. However, label noise in graph data can degrade the generalization performance of the global model. Existing federated label noise learning methods, primarily focused on computer vision tasks, often yield suboptimal results when directly applied to FGL. To address this issue, we propose a robust federated graph learning method with label noise, termed **FedRGL**. Specifically, FedRGL leverages the globally aggregated model and local subgraph structural information to implement a dual-perspective consistency noise-node filtering mechanism under class-aware dynamic thresholds. The resulting *class-aware dual-consistency filtering (CADF)* can also serve as a plug-and-play module, enhancing noise robustness across various subgraph federated learning frameworks. To better exploit the supervisory information from filtered noisy nodes, we employ the natural augmentation techniques from graph contrastive learning to assign high-confidence pseudo-labels to the noise nodes. Additionally, we measure model quality via the average predictive entropy of unlabeled nodes, enabling adaptive robust aggregation on the server side. Extensive experiments on real-world graph datasets show that FedRGL consistently outperforms existing methods under different noise rates, noise types, and client scales, achieving on average **5–8%** higher accuracy and up to **30%** improvement over the weakest baselines under noisy conditions. The source

code is available at `https://github.com/LDer66/FEDRGL-ICML26`.

## 1. Introduction

Graphs are widely used in modeling complex systems due to their ability to visually represent relational information between different entities (Bang et al., 2023; Yang et al., 2023). Graph neural networks (**GNNs**) (Defferrard et al., 2016; Kipf & Welling, 2016), as a promising method for graph information mining, have achieved excellent performance in downstream tasks at node level (Zhu et al., 2020a; Fu et al., 2023; Zhang et al., 2025), edge level (Cai et al., 2021; Yuan et al., 2024), and graph level (Sun et al., 2022; Ju et al., 2023). However, most of the existing GNNs training based on centralized data storage is not applicable to real-world scenarios due to data privacy protection as well as copyright constraints. Therefore, Federated Graph Learning (**FGL**), which combines graph learning and federated learning (Fang et al., 2023; Tan et al., 2025a; Yu et al., 2025), has been proposed to enable joint modeling between clients under the protection of private graph data not going out of the local area.

Current academic research on FGL primarily focuses on optimizing the performance of global or personalized models under Non-independent and identically distributed (***Non-iid***) graph data (Xie et al., 2021; Huang et al., 2023; Tan et al., 2023; Li et al., 2023a; Zhu et al., 2024). For example, FGSSL (Huang et al., 2023) eliminates node semantic bias and subgraph structural bias by leveraging supervised graph contrastive learning and structural relation distillation. FedTAD (Zhu et al., 2024) introduces topology-aware knowledge distillation to transfer the knowledge of local models to the server, achieving an optimal global model. FedSPA (Tan et al., 2025b) further mitigates homophily heterogeneity in federated graph learning via feature propagation decoupling and bias-driven aggregation. However, existing FGL methods assume that local graph data on clients are jointly trained under the premise of clean labels, neglecting the impact of noisy labels in client graph data on the model optimization process, as illustrated in Fig. 1 ①-②. Therefore, this paper further investigates the robustness of federated graph learning under noisy labels, with a particular focus on learning a globally generalizable model from clients holding partial

[1]Key Lab of Education Blockchain and Intelligent Technology, Ministry of Education, Guangxi Normal University [2]School of Physical Science and Technology, Guangxi Normal University [3]School of Computer Science, Big Data and Software, Fuzhou University [4]Guangxi Key Laboratory of Multi-Source Information Mining and Security, Guangxi Normal University. Correspondence to: Chunpei Li <licp@gxnu.edu.cn>.

*Proceedings of the 43rd International Conference on Machine Learning*, Seoul, South Korea. PMLR 306, 2026. Copyright 2026 by the author(s).

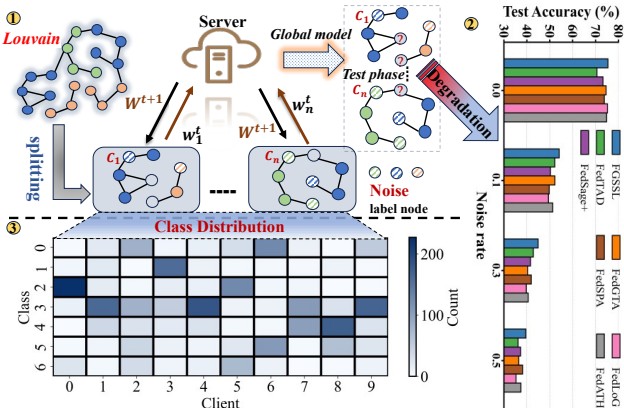

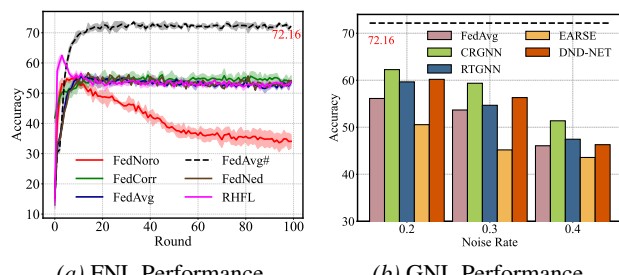

*(a)* FNL Performance      *(b)* GNL Performance

*Figure 2.* (a): Global test accuracy of different FNL methods on the Cora dataset, divided into 10 clients using the Louvain algorithm under a noise rate of 0.3 in pair label noise. FedAvg# represents the FedAvg method trained with clean labels. (b): Global test accuracy of different GNL methods on the Cora dataset, divided into ten clients, under three noise rates of pair label noise.

*Figure 1.* ① Example of Federated Graph Learning with noisy labels. Subgraph data is divided among clients using the Louvain algorithm (Blondel et al., 2008), with varying noise levels. ② Testing accuracy of Seven FGL algorithms on Cora under **uniform** label noise at different rates (*10 clients*), showing lack of robustness to label noise. ③ The class distribution of 10 clients on the Cora dataset. Please zoom in for better visibility.

subgraphs with label noise in node classification tasks.

Existing efforts to mitigate the noisy label problem in federated learning (**FNL**) (Xu et al., 2022; Fang & Ye, 2022; Wu et al., 2023; Li et al., 2024b; Lu et al., 2024) primarily focus on the field of computer vision (CV). However, directly migrating these methods to FGL fails to yield significant performance gains. The primary reasons are the structural heterogeneity between federated subgraphs caused by data partitioning across clients and the class-homogeneity disparity within local subgraphs (*i.e.,* the uneven class distribution across clients, as illustrated in Fig. 1 ③). These challenges result in differing learning rates for nodes of different classes, increasing the difficulty of identifying and managing noisy nodes under the global model's assistance. Current FNL methods employ uniform thresholds for selecting small-loss samples (*e.g.,* FedCorr (Xu et al., 2022)), modify local objective loss functions (*e.g.,* RHFL (Fang & Ye, 2022) and FedNoRo (Wu et al., 2023)), or optimize the global model via negative knowledge distillation (*e.g.,* FedNed). However, these approaches fail to achieve fine-grained noisy node processing within local subgraphs, resulting in limited performance gains. As shown in Fig. 2(a), existing FNL methods fail to effectively mitigate the impact of noisy labels in federated subgraphs.

Furthermore, current research addressing noisy labels in graph learning (**GNL**) primarily focuses on centralized scenarios, including graph semi-supervised node classification (Dai et al.; Qian et al., 2023; Liu et al., 2023; Chen et al., 2024; Xia et al., 2023; Li et al., 2024c), graph classification (Yin et al., 2023; Li et al., 2024a), and graph transfer learning (Yuan et al., 2023a). These methods assume access to complete graph structural information to manage noise effectively. However, in FGL, clients only possess partial sub-

graph information, and the incompleteness of the subgraph structure combined with fewer training nodes (*compared to full graph information*) prevents existing graph noise learning methods from effectively mitigating the impact of noisy labels. Fig. 2(b) further provides the corresponding experimental validation. *Therefore, research on subgraph federated learning with noisy labels is crucial to enhancing the robustness of federated graph learning, offering insights and improvements to better apply FGL techniques to real-world scenarios.*

In this paper, we propose FedRGL, a Federated Graph Learning method that combines global model information and subgraph structural information to mitigate the impact of noisy labels on the global model. ***On the client side***, unlike previous methods that set a uniform threshold and use the Gaussian Mixture Model (GMM) to fit sample losses (Xu et al., 2022), FedRGL addresses the heterogeneity issue of class distribution in local clients. This heterogeneity results in different learning rates of class knowledge for local GNN models (Zhu et al., 2024). FedRGL utilizes both the global model and the corrected local subgraph labels through label propagation to compute sample loss values. Under the constraint of class-aware dynamic thresholds, it proposes a dual-perspective consistency approach for precise noisy node selection(CADF). Importantly, CADF can serve as a plug-and-play strategy that can be coupled with most subgraph federated learning frameworks to enhance their robustness against noisy labels. To effectively leverage the identified noisy nodes, we introduce graph contrastive learning into the local client training process. By leveraging its inherent graph augmentation perspective, high-confidence pseudo-labels are assigned to the noisy nodes, further enhancing the supervision information for model training. Finally, ***On the server side***, the ability of clients to process noisy nodes varies. Therefore, when aggregating the global model, the quality of the models uploaded by the clients must be considered. To this end, FedRGL exploits the characteristics of transductive training in local

subgraphs [1]. Before uploading the model to the server, we compute the predictive entropy on the unlabeled nodes of the local subgraph to assess the model quality during the current training round. The predictive entropy results, along with the model parameters, are uploaded to the server to enable robust global model aggregation. The contributions of our work are summarized as follows:

***Significant Research***: This work investigates the compounding impact of label noise within the intrinsic subgraph FL setting, where class imbalance and the lack of global structure are naturally present. Our study provides crucial insights into robust global model generalization when facing these co-existing adverse conditions.

***New Method***: We propose FedRGL, a label noise learning approach integrating global model knowledge with local subgraph structural knowledge. By considering class distribution and local subgraph training characteristics, FedRGL enables precise noisy node selection and robust global model aggregation. Its core module, the Class-Aware Dual-consistency Filtering (CADF), is designed as a plug-and-play component that can be coupled with most subgraph federated learning frameworks to further improve robustness against noisy labels.

***State-of-the-art Performance***: FedRGL achieves superior test accuracy across multiple real-world graph datasets, consistently outperforming various baselines under different noise types, noise rates, and client numbers.

## 2. Related Work

In this section, we briefly introduce federated graph learning, federated learning with Label Noise, and graph neural networks with Label Noise. Due to space limitations, more detailed descriptions are provided in Appendix A.

**Federated Graph Learning.** Existing Federated Graph Learning (FGL) methods can be broadly categorized into *graph-level FL* and *subgraph-level FL* based on the learning task. In graph-level FL, each client holds multiple independent graphs and applies personalized mining methods, such as GCFL+ (Xie et al., 2021) with gradient-based clustering and FedStar (Tan et al., 2023) with feature-structure decoupling. In contrast, subgraph-level FL assumes that each client holds a partition of a single global graph, and the goal is to perform node-level classification via collaborative training. For global model optimization, representative works like FedSage+ (Zhang et al., 2021) and FedNI (Peng et al., 2022) recover missing structural dependencies using neighbor generators; FGSSL (Huang et al., 2023) and FedTAD (Zhu et al., 2024) further enhance global represen-

tation via contrastive learning and topology-aware knowledge distillation, respectively. Personalized subgraph FL methods such as FedPub (Baek et al., 2023) and FedEgo (Zhang et al., 2023) focus on client-specific adaptation using pseudo-graphs or mixup strategies. Different from these works, this paper investigates the robustness of global subgraph FL models against inter-client structural heterogeneity and intra-client label noise.

**Federated Learning with Label Noise.** Existing FNL methods are mainly categorized into loss-level and sample-level approaches. ❶ *Loss-level*: RHFL (Fang & Ye, 2022) introduces symmetric robust loss and KL divergence constraints on public data, while FedNoro (Wu et al., 2023) and FedNed (Lu et al., 2024) utilize knowledge distillation with aggregation or negative feedback strategies to enhance global robustness against noisy clients. ❷ *Sample-level*: FedCorr (Xu et al., 2022) filters noisy samples via local GMMs, whereas FedDiv (Li et al., 2024b) constructs global GMMs collaboratively for more accurate selection. FedFixer (Ji et al., 2024) alternates noisy sample filtering between global and personalized models. However, these methods are mainly designed for vision tasks and overlook structural heterogeneity inherent in subgraph federated learning, limiting their effectiveness when directly applied.

**Graph Neural Networks with Label Noise.** Existing studies on noisy labels in GNNs focus on centralized settings and can be grouped into three categories: *graph structure augmentation*, *graph contrastive learning*, and *label propagation-based methods*. Structure augmentation methods like NRGNN (Dai et al.) and RTGNN (Qian et al., 2023) connect labeled and unlabeled nodes to improve message passing, with RTGNN further employing a dual-network design to reduce error accumulation. Contrastive methods such as CRGNN (Li et al., 2024c) and CGNN (Yuan et al., 2023b) enhance robustness via dynamic loss, consistency learning, and label correction. Label propagation-based approaches including GNN-cleaner (Xia et al., 2023), $R^2LP$ (Cheng et al., 2024), ERASE (Chen et al., 2024), and DND-NET (Ding et al., 2024) exploit pseudo-label refinement and error-tolerant representation learning. However, these methods rely on complete graph structures and sufficient labeled nodes, which do not hold in subgraph federated learning. As a result, they fail to address the robustness challenges posed by structural heterogeneity and limited supervision in federated settings.

> ***To further show FedRGL's superiority, we compare our approach with existing methods in Appendix B.***

## 3. Methodology

### 3.1. Preliminaries

In subgraph FGL, there exist $M$ clients that collaboratively train a global model with a central server. The $m$-th client

---

[1]Transductive training assumes that all node feature and structure information is known during training, but only the labels of training nodes are available

possesses part of the global graph $G_g = (V_g, A_g, X_g, Y_g)$, represented as a subgraph $G_m = (V_m, A_m, X_m, Y_m)$, where $V_m$ contains training nodes $V_m^{T_r}$, validation nodes $V_m^{V_a}$, and test nodes $V_m^{T_e}$. Each node $v_i \in V_m$ has a feature vector $x_i^m (x_i^m \subseteq X_m)$ and a label $y_i^m (y_i^m \subseteq Y_m)$, where $y_i^m$ may be noisy. In this paper, we assume that the label noise rate among clients varies and is obtained through uniform sampling $U(\eta^l, \eta^u)$ (Xu et al., 2022). FedAvg (McMahan et al., 2017), as the baseline model, serves as the foundation for the FedRGL method proposed in this paper. Each client performs local GNN model training by minimizing the loss function $\arg\min \mathcal{L}_m(w_m; \mathbf{G}_m)$. Then uploads the model parameters $w_m$ along with the number of nodes $|V_m|$ to the central server. The central server performs a federated weighted average to obtain the global model $w = \sum_{m=1}^{M} \frac{|\mathcal{V}_m|}{|\mathcal{V}|} w_m$. Finally, after multiple rounds of collaborative training, the global model will be sent to each client for testing tasks.

## 3.2. FedRGL Method

**Overview**. The FedRGL method, as shown in Fig. 3, operates with distinct client-side and server-side processes. *Client-side*, each client computes the training loss using both the global model and local structure, filtering noisy nodes with a class-aware dynamic threshold (CADF). A graph contrastive approach then assigns high-confidence pseudo-labels to the filtered nodes, improving the quality of the training data. After local training, the clients calculate and upload the prediction entropy of unlabeled nodes to the server. *Server-side*, the server performs aggregation reweighting based on the clients' prediction entropy to achieve a robust global model, which is then broadcasted back to the clients for the next training round. This process repeats until the training is complete.

### 3.2.1. CLASS-AWARE DUAL CONSISTENCY LABEL NOISE FILTERING

❶ **Motivation.** In subgraph FL, structural and distributional heterogeneity across clients makes it difficult to distinguish noisy nodes, as loss patterns vary by both structure and class (see Appendix C for detailed analysis). Existing methods with unified thresholds or single-view filtering fail to adapt to such variations, often confusing hard but correct samples with noisy ones, resulting in degraded robustness (as validated in Fig. 2). To overcome this, we propose a dual-consistency filtering framework combining global model and local structural views, equipped with class-aware dynamic thresholds for adaptive noise detection.

**Global Model View**. Before the $t$-th training round, the $m$-th client uses the global model $W^t$ to calculate the cross-entropy loss $L(G_m)$ for each training node as $\langle \mathcal{L}v_1, \mathcal{L}v_2, \ldots, \mathcal{L}v_k \rangle$. Unlike methods that use a GMM to

fit loss values and set a unified clean probability threshold, our approach considers class distribution differences and intra-class homophily in client subgraphs. As different node categories are learned at varying rates, a unified global threshold is ineffective. Instead, we propose class-aware dynamic thresholds, calculating the mean $t_m^c$ and standard deviation $\sigma_m^c$ of the cross-entropy loss for each class:

$$V_m^c = \{v_i \mid y_i^m = c\}, t_m^c = \frac{1}{|V_m^c|} \sum_{v_i \in V_m^c} \mathcal{L}_{v_i} \quad (1)$$

$$\sigma_m^c = \sqrt{\mathrm{var}(\mathcal{L}_{v_i})}, v_i \in V_m^c \quad (2)$$

$$\rho_m^c = t_m^c + \varphi_1 \sigma_m^c \quad (3)$$

where $V_m^c$ represents the set of nodes in the $m$-th client that belong to class $c$, $\rho_m^c$ is the noise node filtering threshold for class $c$, and $\varphi_1$ is a hyperparameter. Then, according to the dynamic threshold set $\{\rho_m^1, \rho_m^2, \ldots, \rho_m^C\}$ of all classes in the $m$-th client, the clean sample node set $V_m^{C_1}$ under the global model view of the client can be obtained as:

$$V_m^{C_1} = \{v_i \mid y_i^m = c, \mathcal{L}_{v_i} < \rho_m^c\} \quad (4)$$

**Local Structural View**. We introduce the label propagation calculation after global model correction to further enhance the stability and accuracy of noise node filtering. Specifically, to avoid the influence of initial label noise in the local label propagation process, we use the global model $W^t$ to predict the label distribution $P_m = \mathrm{Softmax}(f(V_m^{T_r}, W^t))$ for the training nodes of the $m$-th client at the $t$-th communication round. For the nodes $v_m^{Eq}$ whose predicted labels $\mathrm{argmax}(P_m)$ match their original labels $Y_m$, we retain their original labels during the label propagation process and encode them as one-hot. For the remaining training nodes $V_m^{Re} = V_m^{T_r} \setminus V_m^{Eq}$, the soft labels predicted by the global model are used for initialization.

To avoid the impact of non-training nodes, we employ the subgraph structure masking technique in ERASE (Chen et al., 2024), constructing a masking matrix $MM^T$, where $M \in \{0,1\}^N$, such that the label propagation process only transmits information within the training nodes. The adjacency matrix is denoted by $A_m' = A_m \odot MM^T$. By introducing a non-parametric label propagation algorithm for $k$ steps, we can obtain the class probabilities guided by structural information for each training node:

$$(\hat{Y}_m^{T_r})^k = \alpha(\hat{Y}_m^{T_r})^{k-1} + (1-\alpha)\left(D^{-\frac{1}{2}}A_m'D^{-\frac{1}{2}}\right)(\hat{Y}_m^{T_r})^{k-1} \quad (5)$$

where $D$ is the diagonal matrix, and $\alpha$ is the hyperparameter for label propagation, which is set to 0.6 for all datasets. Particularly, instead of performing label noise processing with hard labels (*i.e.,* one-hot vectors) after label propagation (Chen et al., 2024; Xia et al., 2023), we obtain soft labels (*i.e.,* propagated label scores) here. Then, using the soft

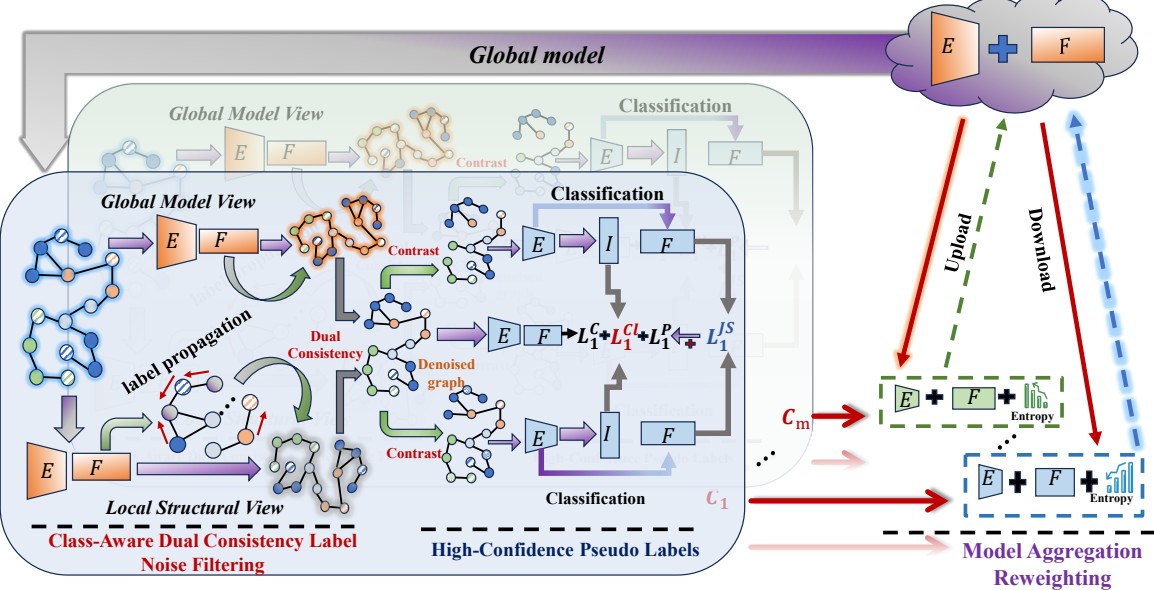

*Figure 3.* The overall framework of FedRGL. On each client, class-aware dual-consistency filtering combines the global model view and the local structural view to identify noisy nodes with class-aware dynamic thresholds. FedRGL then recovers reliable supervision for these nodes through augmented graph views and high-confidence pseudo labels, while optimizing supervised, pseudo-label, consistency, and contrastive objectives. The server further aggregates client models with prediction-entropy-based reweighting, reducing the influence of unreliable updates and producing a more robust global model.

labels after structural propagation and the original labels of the training node $V_m^{T_r}$ to compute the cross-entropy value $\langle \ell_{v_1}, \ell_{v_2}, \ldots, \ell_{v_k} \rangle$, and the set of dynamic thresholds for noise nodes lost by class for the $m$-th client can be obtained as $\{\mu_m^1, \mu_m^2, ..., \mu_m^c\}$, where $\mu_m^c = t_m^c + \varphi_2 \sigma_m^c$. By calculating the dynamic threshold value for each class, we can obtain the clean sample set under the local structural view:

$$V_m^{C_2} = \{v_i | y_m^i = c, \ell_{v_i} < \mu_m^c\} \quad (6)$$

Finally, through the intersection of the clean sample set $V_m^{C_1}$ under the global model view and the clean sample set $V_m^{C_2}$ under the local structural view, we can obtain the final clean sample set $V_m^C = V_m^{C_1} \cap V_m^{C_2}$ and the corresponding noisy node set $V_m^N = V_m^{T_r} \setminus V_m^C$. It is worth noting that to obtain a more optimal and stable performance, we set two hyperparameters $\varphi_1, \varphi_2$ for the dynamic threshold, and the experiment found that setting $\varphi_1 = \varphi_2$ does not significantly degrade the model's generalization ability. As shown in Table 1, the noise filtering accuracy of CADF under 0.5 uniform noise demonstrates its strong and precise capability in handling noisy samples. In addition, ***our noise node selection is executed only once in each communication round, without performing multiple selections along with local epochs.*** ★ To further provide a theoretical characterization of how the proposed class-aware dynamic threshold and dual-consistency mechanism contribute to noise mitigation, we present the following proposition:

**Proposition 3.1** (Noise-selection bound under class-aware

*Table 1.* Noise node filtering success rate of FedRGL on the *Cora* dataset across three different communication rounds and five clients.

| Comm. Rounds | Client 1 | Client 2 | Client 3 | Client 4 | Client 5 |
|---|---|---|---|---|---|
| 30 | 0.843 | 0.815 | 0.773 | 0.805 | 0.791 |
| 50 | 0.895 | 0.857 | 0.844 | 0.856 | 0.849 |
| 70 | 0.931 | 0.918 | 0.887 | 0.904 | 0.892 |

dual-consistency filtering)*. For each client $m$ and class $c$, let the per-class clean and noisy node losses under the global and structural views follow sub-Gaussian distributions with parameters $(\mu_{m,c}^{t(g)}, \sigma_{m,c}^{t(g)})$, $(\mu_{m,c}^{n(g)}, \sigma_{m,c}^{n(g)})$, and $(\mu_{m,c}^{t(s)}, \sigma_{m,c}^{t(s)})$, $(\mu_{m,c}^{n(s)}, \sigma_{m,c}^{n(s)})$, respectively. Define the class-aware thresholds as $\rho_{m,c} = \mu_{m,c}^{t(g)} + \varphi_1 \sigma_{m,c}^{t(g)}$ and $\mu_{m,c} = \mu_{m,c}^{t(s)} + \varphi_2 \sigma_{m,c}^{t(s)}$, and denote by $A_g = \{L_{m,c}^{n(g)} < \rho_{m,c}\}$ and $A_s = \{L_{m,c}^{n(s)} < \mu_{m,c}\}$ the events that a noisy node is misclassified as clean under the global and structural views, respectively. If the separability margins $\Delta_{m,c}^{(g)} = \mu_{m,c}^{n(g)} - \mu_{m,c}^{t(g)} - \varphi_1 \sigma_{m,c}^{t(g)} > 0$ and $\Delta_{m,c}^{(s)} = \mu_{m,c}^{n(s)} - \mu_{m,c}^{t(s)} - \varphi_2 \sigma_{m,c}^{t(s)} > 0$, then the false-selection rates satisfy $\Pr(A_g) \le \exp[-(\Delta_{m,c}^{(g)})^2/2(\sigma_{m,c}^{n(g)})^2]$ and $\Pr(A_s) \le \exp[-(\Delta_{m,c}^{(s)})^2/2(\sigma_{m,c}^{n(s)})^2]$. Under conditional independence between $A_g$ and $A_s$, the intersection event obeys:*

$$\Pr(A_g \cap A_s) \le \exp\left[-\frac{(\Delta_{m,c}^{(g)})^2}{2(\sigma_{m,c}^{n(g)})^2} - \frac{(\Delta_{m,c}^{(s)})^2}{2(\sigma_{m,c}^{n(s)})^2}\right] \quad (7)$$

*This result demonstrates that the class-aware dynamic thresholds adaptively control per-class false selections, while the dual-consistency intersection further achieves exponential contraction of the overall noise inclusion probability. See Appendix D for the detailed proof.*

### 3.2.2. HIGH-CONFIDENCE PSEUDO LABELS

❷ **Motivation.** After CADF filters out potentially noisy nodes, directly ignoring them may weaken local supervision, especially in partial subgraphs with limited labeled nodes and imbalanced classes. To reuse these informative nodes, we incorporate graph contrastive learning (GCL) into local optimization. GCL constructs two enhanced graph views to produce stable predictions for nodes in $V_m^N$, enabling high-confidence pseudo-label assignment and providing additional supervision for local training. Meanwhile, the contrastive objective regularizes the encoder to learn noise-robust representations.

Specifically, in the $t$-th round, the $m$-th client identifies clean nodes $V_m^C$ and noisy nodes $V_m^N$ through label noise filtering. We use edge drop and feature masking techniques from GRACE (Zhu et al., 2020b) to construct two augmented views of the original graph $G_m$, denoted as $G_m^1$ and $G_m^2$, and obtain corresponding embeddings $Z_m^1$ and $Z_m^2$ via the local model's encoder $E_m$ and projection head $I_m$. Additionally, class predictions $p_m^1$ and $p_m^2$ are generated (without passing through $I_m$) using the classifier $F_m$. The noise robustness of the encoder is further enhanced in the embedding space through contrastive loss: $L_m^{Cl} = \frac{1}{2N} \left( L_{cl} \left( Z_m^1, Z_m^2 \right) + L_{cl} \left( Z_m^2, Z_m^1 \right) \right)$, where $L_{cl}$ follows the contrastive loss formulation in GRACE (Zhu et al., 2020b), and its detailed expression is provided in Appendix E. For noisy nodes $V_m^N$, we use the class prediction results from the two augmented views and set a prediction confidence threshold $\gamma$ to obtain the corresponding pseudo-labels:

$$\tilde{Y}_m = \text{Softmax} \left( \left( P_m^1 + P_m^2 \right) / 2 \right) \quad (8)$$

$$V_m^P = \left\{ \left( v_m^i, y_m^i \right) \mid \max \left( y_m^i \right) > \gamma, v_m^i \in V_m^N \right\} \quad (9)$$

where $y_m^i \in \tilde{Y}_m$. Therefore, the $m$-th client uses high-confidence pseudo-labels to increase the local supervision loss information $L_m^P$ as:

$$L_m^P = \frac{\sum_{v_m^i \in V_m^P} \left( L_{ce} \left( \hat{P}_m^{1,i}, \hat{y}_m^i \right) + L_{ce} \left( \hat{P}_m^{2,i}, \hat{y}_m^i \right) \right)}{2} \quad (10)$$

where $L_{ce}$ is the cross-entropy loss, $\hat{P}_m^{1,i} = \text{softmax} \left( P_m^{1,i} \right)$, $\hat{y}_m^i = \text{argmax} \left( y_m^i \right)$. To further enhance the stability of pseudo-labeled nodes during local training, we introduce the Jensen Shannon (JS) divergence for pseudo-labeled nodes $V_m^P$ augmented view predictions $G_m^1, G_m^2$ to ensure consistency with the predictions of the original graph $G_m$. Consequently, the updated loss function for the $m$-th client is

expressed as:

$$L_m = L_m^C + \lambda_{Cl} L_m^{Cl} + \lambda_P L_m^P + \lambda_{Js} L_m^{Js} \left( \hat{P}_m, \hat{P}_m^1, \hat{P}_m^2 \right) \quad (11)$$

where $L_m^C$ denotes the cross-entropy loss for clean nodes and $L_m^{Js}$ represents the JS divergence term (detailed in Appendix F). $\lambda_{Cl}, \lambda_P, \lambda_{Js}$ are hyperparameters. **Notably**, pseudo-labels of noisy nodes are used only within the current epoch to avoid error accumulation.

### 3.2.3. MODEL AGGREGATION REWEIGHTING

❸ **Motivation.** To mitigate the spread of erroneous knowledge during global model aggregation, we propose using model predictive entropy to assess each client's model quality. While predictive entropy on a public dataset at the server has been validated in the CV domain (Huang et al., 2024), **our approach** does not require an unsupervised public dataset on the server side. Leveraging the transductive training approach in subgraph FL, each client calculates the predictive entropy of its local unlabeled nodes before uploading the model to the server, enabling robust reweighting.

After the $t$-th round, client $m$ uses its model $w_m^t$ to compute the predictive entropy $\mathcal{H}_m$ on unlabeled nodes $V_m^U$ (*i.e.,* $V_m^{V_a}$ and $V_m^{T_e}$):

$$\left. \begin{array}{l} \hbar \left( v_m^i \right) = \frac{-1}{|C|} \sum_{c \in C} \hat{P}_m^{i,c} \log \hat{P}_m^{i,c} \\ \hat{P}_m^i = \text{Softmax} \left( P_m^i \right) \end{array} \right\} \Rightarrow \mathcal{H}_m = \frac{\sum_{v_m^i \in V_m^U} \hbar \left( v_m^i \right)}{|V_m^U|} \quad (12)$$

where $\hat{P}_m^{i,c}$ represents the predicted value of class $c$ for the $i$-th node in the unlabeled node set $V_m^U$. Next, we explain why the unlabeled nodes $V_m^U$ are used to estimate the quality of the model. In noisy label learning, the model becomes increasingly confident in its predictions during the training process, even for noisy nodes (Li et al., 2023b). Therefore, if the labeled nodes are directly used for predictive entropy calculation, it will lead to an unstable estimation of model quality. In the model aggregation phase of the $t + 1$ communication round, the server collects the model parameters $\{w_i^t\}_{i=1}^M$ and predictive entropies $\{\mathcal{H}_i\}_{i=1}^M$ from the clients. The server then aggregates these models using the predictive entropies to obtain a robust global model $W^{t+1}$:

$$W^{t+1} = \sum_{m=1}^M \frac{H_m}{\sum_{m=1}^M H_m} w_m^t, \quad H_m = \frac{1}{\mathcal{H}_m + \epsilon} \quad (13)$$

where $H_m$ is the inverse of the predictive entropy of the $m$-th client, and $\epsilon$ is a small constant to avoid division by zero, set to $\epsilon = 1e - 9$ in this work. Additionally, to ensure a certain degree of reliability for the global model during the initial phase, we conduct $T_{\text{warm}}$ rounds of **warm-up** training,

which only involves standard cross-entropy loss training on training nodes without noisy node filtering and server-side aggregation reweighting. The pseudo-code of the FedRGL algorithm can be viewed in the Appendix G.

> *Detailed complexity analysis, privacy analysis, and limitations are deferred to the Appendix H due to space constraints.*

## 4. Experiment

### 4.1. Experimental Setup

**Datasets.** Our experiments are conducted on eight real-world graph datasets, including three citation network datasets (*i.e.,* Cora, CiteSeer, and PubMed) (Yang et al., 2016), two co-author datasets (*i.e.,* CS and Physics) (Kipf & Welling, 2016), and one user-item dataset (*i.e.,* Photo) (Kipf & Welling, 2016). In addition, to further evaluate the scalability of our approach, we conduct experiments on two large-scale OGB benchmarks, *ogbn-arxiv* and *ogbn-products* (Hu et al., 2020). See Appendix I for dataset details.

**Noise Settings.** In this paper, we assume that the training nodes within each client contain noisy labels with varying noise rates. We consider three types of label noise: *Uniform*, *Pair*, and *Instance-dependent* noise. The first two follow the settings in CRGNN (Li et al., 2024c), while the instance-dependent noise is further introduced to simulate more realistic noise scenarios. Detailed configurations are provided in Appendix J.

**Baseline Methods.** We comprehensively compare FedRGL with global model optimization methods (*i.e.,* FedAvg (McMahan et al., 2017), FGSSL (Huang et al., 2023), FedSPA (Tan et al., 2025b), and FedTAD (Zhu et al., 2024)), FNL methods (*i.e.,* FedCorr (Xu et al., 2022), FedNoro (Wu et al., 2023), FedNed (Lu et al., 2024), and RHFL (Fang & Ye, 2022)), as well as GNL methods (*i.e.,* CRGNN (Li et al., 2024c), RTGNN (Qian et al., 2023), ERASE (Chen et al., 2024), and DND-NET (Ding et al., 2024)). Details of the method are in Appendix K.

**Implementation Details.** All clients are optimized using SGD with a learning rate of $1 \times 10^{-2}$ and weight decay $5 \times 10^{-4}$. The total number of communication rounds is set to 100, with 3 local epochs per round. Hyperparameters of FedRGL are tuned using Optuna (Akiba et al., 2019). More detailed settings, including hyperparameter ranges and optimal configurations for each dataset, are provided in Appendix L and in the released code.

### 4.2. Experimental Results and Analysis

**Generalization Performance.** The classification results of FedRGL and state-of-the-art methods under different label-

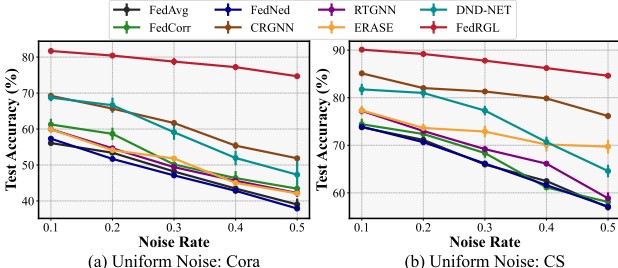

(a) Uniform Noise: Cora  (b) Uniform Noise: CS
*Figure 4.* Effect of Different Noise Scales.

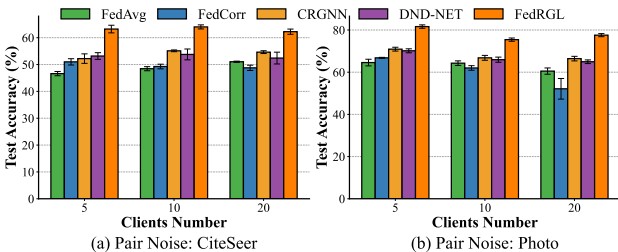

(a) Pair Noise: CiteSeer  (b) Pair Noise: Photo
*Figure 5.* Results under Varying Numbers of Clients.

noise settings are reported in Tab. 2. FedRGL achieves the best test accuracy on most datasets and noise types, showing strong generalization ability in noisy subgraph federated learning. Compared with standard federated graph learning methods such as FedAvg, FGSSL, FedTAD, and FedSPA, FedRGL obtains clear gains because these methods mainly focus on clean-label federated graph learning and do not explicitly handle unreliable labels or noisy client updates. FedRGL also outperforms federated noisy-label learning methods such as FedCorr, FedNoro, FedNed, and RHFL, which are mainly designed for federated visual classification and usually treat samples as independent instances. Thus, they do not explicitly exploit graph topology or model noise propagation over local subgraphs, making them less effective under incomplete local structures, class imbalance, heterogeneous client topology, and graph-dependent label noise.

Under Uniform and Pair noise, the superiority of FedRGL becomes more evident. Although centralized graph noisy-label learning methods such as CRGNN, RTGNN, DND-NET, and ERASE can exploit graph structures, they are not tailored to the federated subgraph scenario, where each client only observes a partial graph with local structural bias. FedRGL addresses these challenges by jointly using the global model view and the local structural view to filter noisy nodes with class-aware dynamic thresholds, recovering reliable supervision through dual-view high-confidence pseudo-labeling, and reducing the influence of unreliable clients through predictive-entropy-based aggregation. These mechanisms alleviate noisy-label memorization at the client side and suppress low-quality updates at the server side, leading to consistently strong results across datasets, noise types, and client numbers. Under clean-label settings (*i.e.,*

*Table 2.* Comparison of the accuracy of state-of-the-art methods at a noise rate of $\eta = 0.3$ on six graph datasets, where **Normal** denotes clean labeling. The best precision is denoted by ( ▭ ).

| Dataset | Cora (5 Clients) | | | CiteSeer (5 Clients) | | | PubMed (10 Clients) | | |
|---|---|---|---|---|---|---|---|---|---|
| Method | Normal | Uniform | Pair | Normal | Uniform | Pair | Normal | Uniform | Pair |
| FedAvg | $81.04_{\pm0.37}$ | $48.19_{\pm0.34}$ | $55.80_{\pm0.18}$ | $71.12_{\pm0.44}$ | $38.73_{\pm0.84}$ | $46.62_{\pm0.79}$ | $85.86_{\pm0.07}$ | $74.36_{\pm0.45}$ | $71.21_{\pm0.28}$ |
| FGSSL | $82.16_{\pm0.21}$ | $51.81_{\pm0.56}$ | $57.58_{\pm0.85}$ | $71.88_{\pm0.35}$ | $43.27_{\pm0.88}$ | $49.78_{\pm1.24}$ | $86.44_{\pm0.15}$ | $77.57_{\pm0.11}$ | $73.07_{\pm0.17}$ |
| FedTAD | $81.82_{\pm1.01}$ | $47.20_{\pm1.26}$ | $55.40_{\pm0.37}$ | $70.41_{\pm1.09}$ | $39.35_{\pm0.29}$ | $45.98_{\pm0.80}$ | $85.65_{\pm0.05}$ | $73.46_{\pm0.66}$ | $72.11_{\pm0.19}$ |
| FedSPA | $82.50_{\pm0.42}$ | $50.39_{\pm0.79}$ | $57.38_{\pm0.65}$ | $71.95_{\pm0.33}$ | $40.77_{\pm0.52}$ | $48.56_{\pm0.78}$ | $86.19_{\pm0.42}$ | $74.73_{\pm0.59}$ | $70.36_{\pm0.66}$ |
| FedCorr | $79.05_{\pm0.82}$ | $50.14_{\pm0.37}$ | $55.16_{\pm0.97}$ | $71.42_{\pm0.32}$ | $45.16_{\pm0.91}$ | $51.00_{\pm1.17}$ | $81.18_{\pm0.61}$ | $75.04_{\pm0.67}$ | $70.90_{\pm2.41}$ |
| FedNoro | $81.19_{\pm0.26}$ | $45.54_{\pm1.02}$ | $45.80_{\pm2.52}$ | $69.99_{\pm0.63}$ | $37.53_{\pm0.64}$ | $43.71_{\pm0.26}$ | $84.10_{\pm0.16}$ | $72.37_{\pm0.34}$ | $65.99_{\pm0.04}$ |
| FedNed | $80.68_{\pm0.00}$ | $47.11_{\pm0.04}$ | $56.85_{\pm1.02}$ | $69.50_{\pm0.71}$ | $37.45_{\pm0.06}$ | $47.90_{\pm0.07}$ | $85.76_{\pm0.18}$ | $74.44_{\pm0.63}$ | $70.83_{\pm0.03}$ |
| RHFL | $80.17_{\pm0.47}$ | $49.92_{\pm0.22}$ | $53.36_{\pm0.99}$ | $66.05_{\pm1.11}$ | $39.80_{\pm1.04}$ | $47.88_{\pm0.39}$ | $84.25_{\pm0.07}$ | $61.87_{\pm0.81}$ | $59.70_{\pm0.49}$ |
| CRGNN | $83.09_{\pm0.87}$ | $61.67_{\pm0.80}$ | $63.24_{\pm0.74}$ | $71.06_{\pm0.59}$ | $48.09_{\pm1.00}$ | $52.21_{\pm1.77}$ | $84.70_{\pm0.11}$ | $78.28_{\pm0.31}$ | $73.46_{\pm0.44}$ |
| RTGNN | $80.65_{\pm0.24}$ | $49.40_{\pm0.19}$ | $57.50_{\pm0.79}$ | $71.08_{\pm0.43}$ | $39.45_{\pm0.39}$ | $48.13_{\pm0.54}$ | $85.09_{\pm0.06}$ | $77.80_{\pm0.76}$ | $73.04_{\pm0.36}$ |
| DND-NET | $81.54_{\pm0.87}$ | $59.11_{\pm2.11}$ | $54.68_{\pm1.36}$ | $70.89_{\pm0.41}$ | $48.63_{\pm0.94}$ | $55.04_{\pm1.25}$ | $84.91_{\pm0.59}$ | $75.62_{\pm0.98}$ | $72.33_{\pm1.64}$ |
| ERASE | $71.40_{\pm0.16}$ | $51.81_{\pm0.63}$ | $53.34_{\pm0.82}$ | $63.42_{\pm0.80}$ | $44.68_{\pm0.75}$ | $47.76_{\pm3.90}$ | $76.98_{\pm0.36}$ | $67.00_{\pm0.99}$ | $61.69_{\pm1.51}$ |
| **FedRGL** | $82.12_{\pm0.51}$ | $78.75_{\pm0.98}$ | $75.14_{\pm0.30}$ | $71.52_{\pm0.14}$ | $66.08_{\pm1.02}$ | $63.22_{\pm0.82}$ | $85.33_{\pm0.12}$ | $81.77_{\pm0.32}$ | $76.84_{\pm0.23}$ |
| Dataset | CS (10 Clients) | | | Photo (20 Clients) | | | Physics (20 Clients) | | |
| Method | Normal | Uniform | Pair | Normal | Uniform | Pair | Normal | Uniform | Pair |
| FedAvg | $86.95_{\pm0.09}$ | $65.95_{\pm0.21}$ | $68.95_{\pm0.60}$ | $85.88_{\pm0.18}$ | $72.55_{\pm1.15}$ | $60.52_{\pm1.49}$ | $92.78_{\pm0.19}$ | $69.05_{\pm0.26}$ | $70.96_{\pm0.98}$ |
| FGSSL | $88.75_{\pm0.11}$ | $69.67_{\pm0.37}$ | $71.19_{\pm1.07}$ | $83.92_{\pm0.91}$ | $70.99_{\pm1.14}$ | $64.31_{\pm1.77}$ | $93.46_{\pm0.14}$ | $78.25_{\pm0.34}$ | $76.52_{\pm0.85}$ |
| FedTAD | $86.33_{\pm0.03}$ | $72.22_{\pm0.26}$ | $70.97_{\pm1.91}$ | $85.20_{\pm1.05}$ | $70.15_{\pm1.48}$ | $55.64_{\pm6.82}$ | $93.12_{\pm0.39}$ | $70.29_{\pm0.56}$ | $70.06_{\pm0.59}$ |
| FedSPA | $88.73_{\pm0.30}$ | $68.82_{\pm0.45}$ | $69.75_{\pm0.55}$ | $87.28_{\pm0.42}$ | $70.29_{\pm0.85}$ | $63.37_{\pm0.66}$ | $92.93_{\pm0.44}$ | $72.43_{\pm0.52}$ | $69.85_{\pm0.77}$ |
| FedCorr | $87.49_{\pm0.14}$ | $68.37_{\pm1.01}$ | $70.14_{\pm1.26}$ | $78.42_{\pm3.32}$ | $62.62_{\pm2.39}$ | $52.13_{\pm4.89}$ | $93.19_{\pm0.26}$ | $75.36_{\pm0.63}$ | $78.57_{\pm0.73}$ |
| FedNoro | $82.87_{\pm0.18}$ | $55.15_{\pm0.44}$ | $62.42_{\pm2.99}$ | $76.82_{\pm0.01}$ | $48.51_{\pm1.32}$ | $45.15_{\pm0.34}$ | $91.36_{\pm0.69}$ | $57.65_{\pm1.10}$ | $55.50_{\pm2.36}$ |
| FedNed | $87.50_{\pm0.04}$ | $66.18_{\pm0.13}$ | $66.55_{\pm0.76}$ | $83.37_{\pm0.36}$ | $66.42_{\pm1.81}$ | $63.66_{\pm1.06}$ | $92.79_{\pm0.06}$ | $68.80_{\pm0.14}$ | $72.37_{\pm0.78}$ |
| RHFL | $87.27_{\pm0.20}$ | $56.13_{\pm0.38}$ | $56.62_{\pm1.93}$ | $83.42_{\pm0.05}$ | $66.67_{\pm1.67}$ | $66.38_{\pm2.38}$ | $93.18_{\pm0.02}$ | $58.22_{\pm0.19}$ | $65.58_{\pm1.75}$ |
| CRGNN | $87.63_{\pm0.73}$ | $81.30_{\pm0.54}$ | $74.56_{\pm1.41}$ | $85.09_{\pm0.27}$ | $71.66_{\pm1.05}$ | $66.44_{\pm1.08}$ | $92.87_{\pm0.12}$ | $83.37_{\pm0.17}$ | $80.85_{\pm0.47}$ |
| RTGNN | $84.73_{\pm0.29}$ | $69.19_{\pm0.57}$ | $70.12_{\pm0.21}$ | $82.35_{\pm0.31}$ | $73.86_{\pm1.41}$ | $67.44_{\pm0.95}$ | $91.89_{\pm0.24}$ | $74.35_{\pm3.08}$ | $75.21_{\pm1.69}$ |
| DND-NET | $87.22_{\pm0.03}$ | $77.30_{\pm0.98}$ | $67.14_{\pm1.14}$ | $84.76_{\pm0.75}$ | $71.29_{\pm1.33}$ | $64.96_{\pm0.87}$ | $91.81_{\pm0.93}$ | $83.25_{\pm1.08}$ | $76.38_{\pm1.36}$ |
| ERASE | $87.64_{\pm0.21}$ | $72.88_{\pm1.19}$ | $70.08_{\pm1.26}$ | $81.40_{\pm0.34}$ | $53.94_{\pm3.64}$ | $50.76_{\pm1.39}$ | $90.53_{\pm0.59}$ | $78.90_{\pm0.87}$ | $70.42_{\pm1.16}$ |
| **FedRGL** | $90.58_{\pm0.13}$ | $87.79_{\pm0.22}$ | $81.01_{\pm0.32}$ | $88.42_{\pm0.27}$ | $84.06_{\pm1.13}$ | $77.56_{\pm0.73}$ | $93.62_{\pm0.11}$ | $89.66_{\pm0.53}$ | $86.39_{\pm0.51}$ |

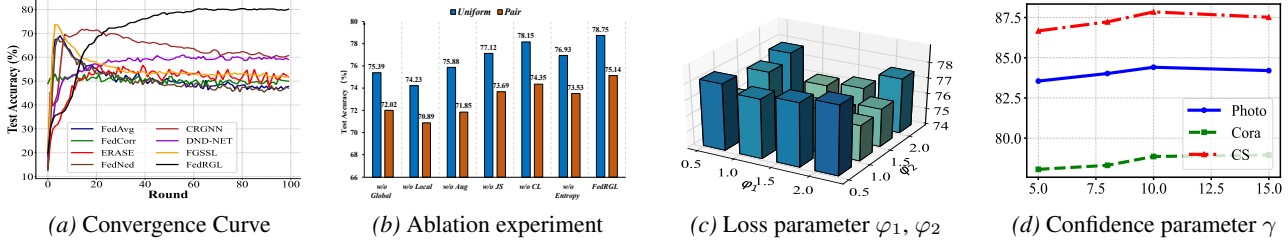

*(a)* Convergence Curve     *(b)* Ablation experiment     *(c)* Loss parameter $\varphi_1$, $\varphi_2$     *(d)* Confidence parameter $\gamma$

*Figure 6.* (a) Convergence curves of different methods on the Cora dataset under uniform label noise. (b) Ablation study of key components in FedRGL under uniform and pair noise, details of each component are in Appendix O. (c) Impact of loss parameters $\varphi_1$ and $\varphi_2$ on model performance. (d) Effect of the high-confidence pseudo-label threshold $\gamma$ across Cora, CS, and Photo datasets.

Normal), FedRGL also maintains competitive or superior results, since incomplete local structures, class imbalance, and heterogeneous client distributions still make local training signals and client updates uneven in quality. Additional validation under the more challenging instance-dependent noise setting is provided in Appendix M, further confirming the robustness of FedRGL beyond uniform and pair label noise.

**Different noise scales and client numbers.** We evaluate the performance of FedRGL under varying noise rates and client numbers. As shown in Fig. 4, FedRGL consistently outperforms baseline methods on *Cora* (5 clients) and *CS* (10 clients) under uniform noise. Notably, the performance gap widens as the noise level increases, and FedRGL maintains high accuracy even under severe noise, demonstrating strong robustness. Fig. 5 presents results on *CiteSeer* and

*Photo* with 5, 10, and 20 clients under a fixed pair noise rate of 0.3. FedRGL achieves the best accuracy across all settings, showing superior scalability and stability compared to existing methods. While *CRGNN* provides moderate improvements over *FedAvg*, its performance remains limited relative to FedRGL. *See Appendix N for analysis on **noisy client ratios** and **participation rates***.

**Plug-and-play verification of CADF.** To evaluate the general applicability of the proposed CADF module, we integrate it into seven representative subgraph FL frameworks under 0.3 uniform noise, including FGSSL, FedTAD, FedSPA, FedSage+ (Zhang et al., 2021), FedLoG (Kim et al.), FedGTA (Li et al., 2023a), and FedATH (Fu et al., 2025). As shown in Tab. 3, CADF consistently improves the performance of all frameworks across Cora, CS, and Physics, with average gains ranging from 16.65% to 19.55%.

*Table 3.* Performance improvement after applying the proposed CADF module to different subgraph FL frameworks under 0.3 uniform noise.

| Method | Cora | CS | Physics | Improv. (%) |
|--------|------|-----|---------|-------------|
| FGSSL | $51.81 \pm 0.56$ | $69.67 \pm 0.37$ | $78.25 \pm 0.34$ | – |
| **+CADF** | $79.19 \pm 0.63$ | $85.51 \pm 0.32$ | $88.79 \pm 0.71$ | ↑**17.62** |
| FedTAD | $47.20 \pm 1.26$ | $72.22 \pm 0.26$ | $70.29 \pm 0.56$ | – |
| **+CADF** | $76.30 \pm 0.78$ | $85.92 \pm 0.52$ | $87.15 \pm 0.85$ | ↑**19.55** |
| FedSPA | $50.39 \pm 0.79$ | $68.82 \pm 0.45$ | $72.43 \pm 0.52$ | – |
| **+CADF** | $75.55 \pm 0.80$ | $83.64 \pm 0.58$ | $87.05 \pm 0.69$ | ↑**18.20** |
| FedSage+ | $52.76 \pm 0.61$ | $70.32 \pm 0.42$ | $74.88 \pm 0.58$ | – |
| **+CADF** | $77.54 \pm 0.67$ | $85.73 \pm 0.33$ | $87.21 \pm 0.63$ | ↑**17.51** |
| FedLoG | $49.82 \pm 0.74$ | $68.95 \pm 0.56$ | $72.10 \pm 0.64$ | – |
| **+CADF** | $76.12 \pm 0.82$ | $83.44 \pm 0.51$ | $86.87 \pm 0.77$ | ↑**18.52** |
| FedGTA | $53.05 \pm 0.69$ | $72.81 \pm 0.49$ | $76.92 \pm 0.61$ | – |
| **+CADF** | $77.45 \pm 0.70$ | $86.32 \pm 0.46$ | $88.95 \pm 0.58$ | ↑**16.65** |
| FedATH | $50.96 \pm 0.73$ | $69.57 \pm 0.55$ | $73.18 \pm 0.59$ | – |
| **+CADF** | $76.84 \pm 0.77$ | $84.26 \pm 0.53$ | $87.69 \pm 0.65$ | ↑**18.36** |

*Table 4.* Accuracy on large-scale graph datasets under different client counts and noise types.

| ogbn-arxiv | | | | |
|--------|------|------|------|------|
| Method | Uniform (20) | Pair (20) | Uniform (30) | Pair (30) |
| FedAvg | $51.83 \pm 0.07$ | $48.01 \pm 0.29$ | $50.76 \pm 0.21$ | $45.33 \pm 0.05$ |
| FedCorr | $52.73 \pm 0.27$ | $46.83 \pm 0.55$ | $51.59 \pm 0.17$ | $46.27 \pm 0.43$ |
| FedNed | $50.21 \pm 0.53$ | $48.07 \pm 0.27$ | $49.47 \pm 0.21$ | $45.21 \pm 0.03$ |
| RTGNN | $52.31 \pm 0.05$ | $47.33 \pm 0.14$ | $51.31 \pm 0.34$ | $44.08 \pm 0.17$ |
| CRGNN | OOM | OOM | $44.42 \pm 0.56$ | $41.62 \pm 0.36$ |
| **FedRGL** | $\mathbf{56.61 \pm 0.37}$ | $\mathbf{52.21 \pm 0.17}$ | $\mathbf{55.64 \pm 0.25}$ | $\mathbf{50.83 \pm 0.22}$ |

| ogbn-products | | | | |
|--------|------|------|------|------|
| Method | Uniform (40) | Pair (40) | Uniform (50) | Pair (50) |
| FedAvg | $58.35 \pm 0.42$ | $53.72 \pm 0.71$ | $58.02 \pm 0.33$ | $51.59 \pm 0.31$ |
| FedCorr | $58.62 \pm 0.68$ | $51.39 \pm 0.89$ | $57.29 \pm 0.44$ | $51.78 \pm 0.68$ |
| FedNed | $56.39 \pm 1.21$ | $53.96 \pm 0.62$ | $57.15 \pm 0.57$ | $50.06 \pm 0.58$ |
| RTGNN | $59.56 \pm 0.57$ | $54.53 \pm 0.45$ | $58.77 \pm 0.62$ | $52.69 \pm 0.35$ |
| CRGNN | $58.22 \pm 0.52$ | $54.12 \pm 0.66$ | $59.74 \pm 0.73$ | $52.12 \pm 0.55$ |
| **FedRGL** | $\mathbf{63.71 \pm 0.49}$ | $\mathbf{59.83 \pm 0.52}$ | $\mathbf{63.32 \pm 0.59}$ | $\mathbf{57.71 \pm 0.62}$ |

Since CADF mainly acts as a noisy-node filtering module before subsequent local training, it can be flexibly incorporated into different subgraph FL pipelines without changing their core optimization procedures. The consistent gains across diverse frameworks demonstrate its strong plug-and-play capability, generalizability, and robustness against label noise.

**Large-scale Graph Data Results.** To evaluate the scalability of FedRGL on large-scale graphs, we conduct experiments on both *ogbn-arxiv* and *ogbn-products* under a 0.4 noise rate with different client counts and noise types. As shown in Tab. 4, existing federated learning methods and graph noisy-label learning methods generally suffer from noticeable performance degradation on large-scale noisy graphs. This degradation becomes more evident under the more challenging pair-noise setting and when the number of clients increases, where fragmented local subgraphs and heterogeneous client distributions make robust training more difficult. In contrast, FedRGL consistently achieves the best performance across all settings on both datasets. On *ogbn-arxiv*, FedRGL improves over FedAvg by 4.78% and 4.88% under uniform noise with 20 and 30 clients, respectively,

and by 4.20% and 5.50% under pair noise. On the larger *ogbn-products* dataset, FedRGL also brings consistent gains over FedAvg, with improvements of 5.36% and 5.30% under uniform noise and 6.11% and 6.12% under pair noise. These results demonstrate that FedRGL can effectively alleviate noisy-label effects and unreliable client updates in large-scale noisy subgraph federated learning. For training efficiency, we perform noisy node filtering every 10 communication rounds and disable the graph contrastive loss term on both large-scale datasets.

**Diagnostic Analysis.** Fig. 6 provides a comprehensive analysis of FedRGL. ❶ *Stability Analysis:* As shown in Fig. 6(a), FedRGL achieves more stable convergence than other baselines under uniform label noise on Cora. Unlike methods such as CRGNN, which suffer from a performance drop in later rounds, FedRGL maintains a stable upward trend, indicating that noisy-node filtering and robust aggregation help alleviate noisy-label memorization. ❷ *Ablation Experiment:* Fig. 6(b) presents the ablation study under $\eta = 0.3$ on Cora. The results show that both the global model view and local structural view are important for reliable noisy-node filtering, since removing either of them leads to a clear performance drop. The graph augmentation branch also contributes substantially to robust pseudo-label recovery, while the contrastive loss provides an additional but relatively smaller improvement. Moreover, the results validate the benefits of two-view prediction consistency and entropy-based server reweighting, which respectively enhance local supervision reliability and reduce the influence of uncertain client updates. ❸ *Parameter Sensitivity Analysis:* Fig. 6(c) analyzes the effects of $\varphi_1$ and $\varphi_2$, where FedRGL shows stable performance under different parameter choices and generally performs well when $\varphi_1 = \varphi_2$. Fig. 6(d) shows that FedRGL is also insensitive to the confidence threshold $\gamma$ across Cora, CS, and Photo under pair noise. Detailed descriptions of the ablation variants and complete results on other datasets are provided in Appendix O.

## 5. Conclusion

This paper addresses noisy labels in federated subgraph node classification, a problem distinct from conventional vision-based studies. We propose **FedRGL**, a robust federated graph learning framework that tackles both label noise and subgraph heterogeneity. FedRGL introduces a *class-aware dual-consistency filtering* module that fuses global and local perspectives for accurate noisy node identification and can be hot-plugged into existing subgraph FL methods. Moreover, graph contrastive learning assigns high-confidence pseudo-labels, while entropy-based aggregation stabilizes global optimization. Extensive experiments demonstrate the superior robustness, adaptability, and scalability of FedRGL across diverse federated graph learning scenarios.

## Acknowledgements

This work was supported by the Guangxi Natural Science Foundation (No. 2026GXNSFBA00640186 and No. 2025GXNSFBA069295), the National Natural Science Foundation of China (No. U21A20474), and the Innovation Project of Guangxi Graduate Education (No. YCBZ2025078). Supported by the China Postdoctoral Science Foundation under Grant Number 2025MD784113, the Basic Ability Enhancement Program for Young and Middle-aged Teachers of Guangxi under Grant No. 2025KY0105, the Key Laboratory of Education Blockchain and Intelligent Technology, the Ministry of Education, under Grant No. EBME24-02, and the Postdoctoral Fellowship Program of CPSF under Grant Number GZC20251060.

## Impact Statement

This work aims to improve the robustness of federated graph learning under noisy labels, particularly when distributed clients hold partial subgraphs with heterogeneous structures and label distributions. It may benefit privacy-preserving graph learning applications such as recommendation, finance, healthcare, social network analysis, and educational analytics, where centralized data collection is often infeasible and label noise is common. By mitigating noisy labels and unreliable client updates, the proposed method can help build more stable distributed graph learning systems. However, it does not eliminate all risks in federated graph learning. Model updates, predictions, or learned representations may still leak sensitive information without additional privacy-preserving mechanisms, and robustness to label noise does not remove potential biases in graph structures, labels, or client populations. Therefore, deployment in sensitive domains should involve privacy protection, fairness evaluation, security auditing, domain-specific validation, and human oversight for high-stakes decisions.

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

# A. Related Work

## A.1. Federated Graph Learning

Existing Federated Graph Learning (FGL) methods can be broadly classified into two types based on the graph task type: graph FL (Xie et al., 2021; Tan et al., 2023; 2024) and subgraph FL (Zhang et al., 2021; Huang et al., 2023; Zhu et al., 2024; Baek et al., 2023). **In graph FL,** each client holds a set of graph data and designs personalized graph mining methods to accomplish graph-level classification tasks. For example, GCFL+ (Xie et al., 2021) achieves personalized parameter updates through gradient-based dynamic clustering, while FedStar (Tan et al., 2023) proposes a feature and structure decoupled encoder to learn invariant structural knowledge between clients alongside personalized knowledge from local graph data. Unlike graph FL, *subgraph FL* involves each client possessing only partial subgraph knowledge of a complete graph. Collaborative training of a global model or personalized local models is performed to accomplish node-level classification tasks. *For global models,* FedSage+ (Zhang et al., 2021) and FedNI (Peng et al., 2022) use neighbor generators to restore missing inter-client connections and neighboring nodes. FGSSL (Huang et al., 2023) enhances the learning of graph nodes and structural information through supervised graph contrastive learning and relational distillation, while FedTAD (Zhu et al., 2024) improves the generalization performance of global models through topology-aware knowledge distillation without direct data sharing. *For personalized models,* FedPub (Baek et al., 2023) introduces global pseudo-graphs and local personalized parameter masks to optimize client models, while FedEgo (Zhang et al., 2023) employs graph mixup and model separation techniques to train personalized local models. AdaFGL (Li et al., 2024d) proposes a new paradigm for personalized FGL by decoupling collaborative training and feature propagation. Additionally, FEDLIT (Xie et al., 2023) and FGGP (Wan et al., 2024) explore new directions for FGL by addressing heterogeneity in potential link types and graph domain shifts. Unlike the aforementioned studies, this paper investigates the generalization performance of global models from a robustness perspective, focusing on the dual challenges posed by inter-client graph structure heterogeneity and intra-client label noise.

## A.2. Federated Learning with label noise

Current Federated Learning with label noise (FNL) research is mainly divided into methods based on loss-level and sample-level approaches. ❶ *Loss-level.* RHFL (Fang & Ye, 2022) utilizes a symmetric robust loss and KL divergence constrained on a public dataset to train personalized robust models. FedNoro (Wu et al., 2023) employs knowledge distillation and a distance-aware aggregation function to jointly update the noise-robust global model. FedNed (Lu et al., 2024) adopts a negative distillation strategy to mitigate the performance impact of "noisy clients" on the global model and utilizes pseudo-labeling techniques to leverage diverse information from noisy clients. ❷ *Sample-level.* FedCorr (Xu et al., 2022)uses a GMM algorithm to filter out noisy clients and noisy samples, and re-labels the selected samples using the global model. Compared to the use of local GMMs to filter noisy samples, FedDiv (Li et al., 2024b) collaborates with all clients to build global GMM parameters for the precise selection of noisy samples. FedFixer (Ji et al., 2024) introduces a dual collaborative network of personalized and global models to alternately select noisy samples. However, the aforementioned FNL methods are primarily designed for handling noisy labels in federated learning within the vision domain. Directly applying these methods to address label noise in subgraph federated learning fails to consider the heterogeneity of subgraph structures, leading to suboptimal performance improvements.

## A.3. Graph Neural Networks with Label Noise

Existing research on noisy labels in graph neural networks primarily focuses on centralized scenarios, which can be broadly categorized into three types: graph structure augmentation level, graph contrast level, and graph structure propagation level. *Graph structure augmentation level*: NRGNN (Dai et al.) and RTGNN (Qian et al., 2023) both enhance the propagation of graph information by linking labeled and unlabeled nodes, with the difference that the latter employs a dual-network structure to further prevent error accumulation. *Graph contrast level*: Leveraging the robustness of unsupervised graph contrastive methods to label noise, CRGNN (Li et al., 2024c) improves model test performance by using dynamic loss and cross-space consistency, while CGNN (Yuan et al., 2023b) corrects labels using neighbor label information. *Graph structure propagation level*: Benefiting from the efficiency of label propagation algorithms, GNN-cleaner (Xia et al., 2023) and $R^2LP$ (Cheng et al., 2024) address noise in semi-supervised node classification when a certain amount of clean training node label information is available. ERASE (Chen et al., 2024) further relaxes the constraint of clean sample information by learning error-tolerant node representations using prototype pseudo-labels and structure-propagated pseudo-labels. DND-NET (Ding et al., 2024) introduces feature propagation to mitigate the impact of noisy labels on model performance. However, these

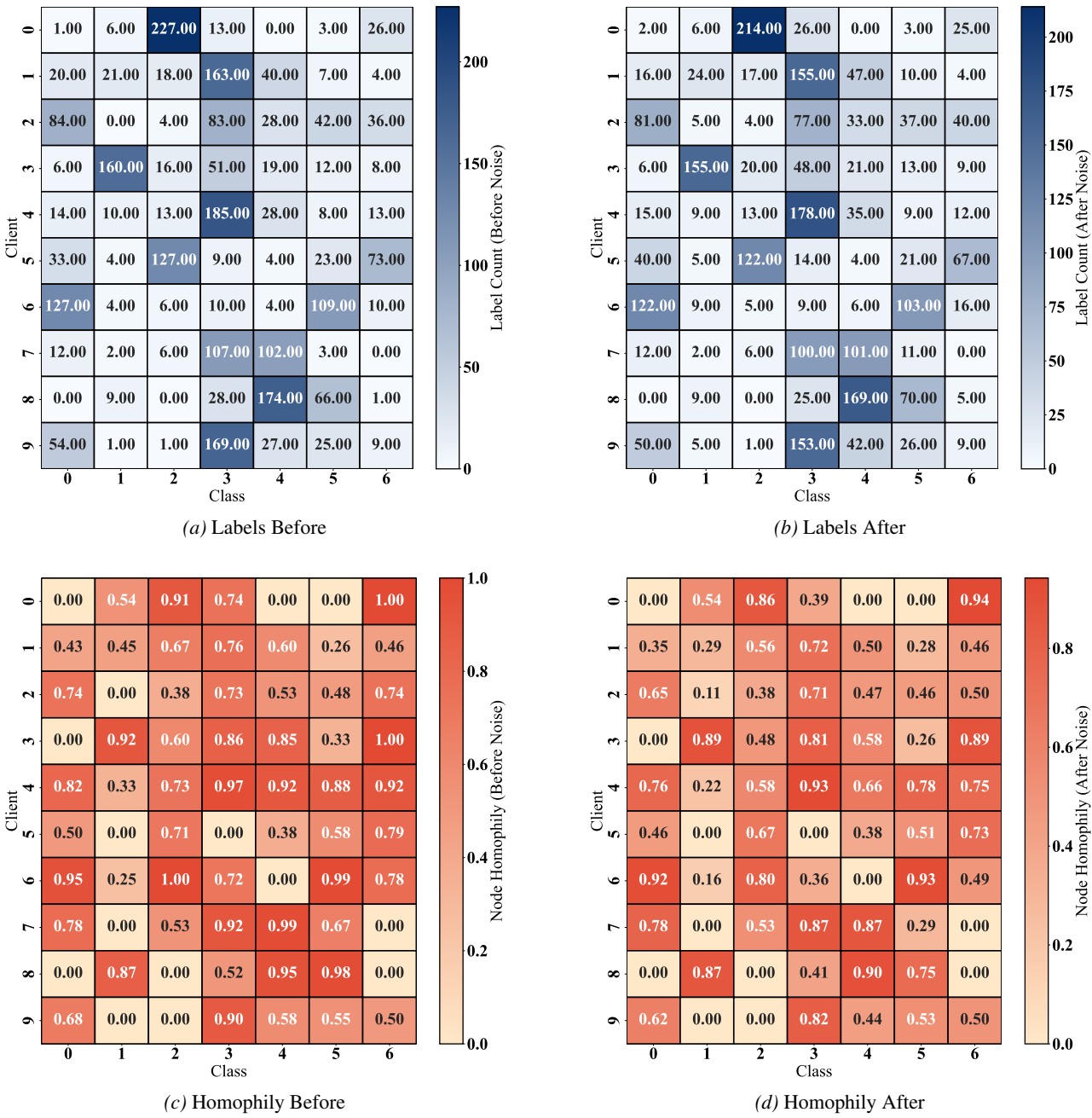

*(a)* Labels Before

*(b)* Labels After

*(c)* Homophily Before

*(d)* Homophily After

*Figure 7.* Client-level label distributions and node homophily on Cora before and after adding label noise.

methods are designed for centralized scenarios, where complete graph structure information and abundant training nodes are available. When applied to subgraph federated learning with noisy labels, the incomplete and heterogeneous subgraph structures, along with the scarcity of training nodes, prevent existing GNL methods from maximizing the robustness of the federated global model.

## B. Comparison with Existing Methods

Unlike existing FNL methods targeting the CV domain, this study focuses on the more complex problem of learning with label noise in federated subgraphs, where the structural heterogeneity of subgraphs and label noise across clients exacerbate the difficulty of filtering noisy nodes, making it challenging to directly apply existing FNL methods. Moreover, FedRGL

leverages the label propagation algorithm to filter noisy nodes within clients, but it differs from existing methods in three key aspects. **First**, our proposed method does not require a prior condition of clean labels, whereas both GNN-cleaner and R$^2$LP rely on the assistance of clean label priors. **Second**, before label propagation, FedRGL uses a robust global model to assess and correct the initial node labels, whereas ERASE directly utilizes the initial node label information. **Lastly**, the existing three methods directly utilize the one-hot label information after label propagation for label correction, while FedRGL computes the loss using the propagated soft labels (e.g., probability distributions) along with the original labels, providing a structural perspective for noisy node filtering.

## C. Verification of Subgraph Heterogeneity Leading to Increased Difficulty in Label Noise Handling

In federated subgraph learning, subgraph heterogeneity among clients (e.g., structural heterogeneity and label heterogeneity caused by data distribution) increases the difficulty of identifying and managing label noise in federated scenarios. As shown in the four subfigures of Fig. 7, the distribution of class labels and homophily in each client, before and after adding pair noise with a rate of 0.3 on the Cora dataset, was analyzed after partitioning the clients using the Louvain algorithm. From the figures, we can observe that the class label distribution and homophily distribution in each client change significantly after adding label noise, especially in terms of homophily distribution. Additionally, it is evident that the label distribution across clients is uneven and sometimes imbalanced, which, combined with the unique structural attributes of the subgraphs, makes it difficult for existing federated label noise learning methods and graph label noise learning methods to achieve effective performance improvements.

## D. Theoretical Analysis and Proofs

**Proof.** Let $A_g = \{L_{m,c}^{n(g)} < \rho_{m,c}\}$ and $A_s = \{L_{m,c}^{n(s)} < \mu_{m,c}\}$ denote the events that a noisy node is misclassified as clean under the global and structural views, respectively. For each class $c$ in client $m$, the clean and noisy node losses under both views are assumed to follow sub-Gaussian distributions with parameters $(\mu_{m,c}^{t(g)}, \sigma_{m,c}^{t(g)})$, $(\mu_{m,c}^{n(g)}, \sigma_{m,c}^{n(g)})$, and $(\mu_{m,c}^{t(s)}, \sigma_{m,c}^{t(s)})$, $(\mu_{m,c}^{n(s)}, \sigma_{m,c}^{n(s)})$. The class-aware dynamic thresholds are defined as $\rho_{m,c} = \mu_{m,c}^{t(g)} + \varphi_1 \sigma_{m,c}^{t(g)}$ and $\mu_{m,c} = \mu_{m,c}^{t(s)} + \varphi_2 \sigma_{m,c}^{t(s)}$.

**(1) Single-view bound.** The separation margins between noisy and clean nodes in each view are given by $\Delta_{m,c}^{(g)} = \mu_{m,c}^{n(g)} - \mu_{m,c}^{t(g)} - \varphi_1 \sigma_{m,c}^{t(g)}$ and $\Delta_{m,c}^{(s)} = \mu_{m,c}^{n(s)} - \mu_{m,c}^{t(s)} - \varphi_2 \sigma_{m,c}^{t(s)}$. According to the one-sided tail inequality of a sub-Gaussian variable,

$$\Pr(A_g) \le \exp\left(-\frac{(\Delta_{m,c}^{(g)})^2}{2(\sigma_{m,c}^{n(g)})^2}\right), \quad \Pr(A_s) \le \exp\left(-\frac{(\Delta_{m,c}^{(s)})^2}{2(\sigma_{m,c}^{n(s)})^2}\right) \tag{14}$$

These inequalities indicate that, for sufficiently positive separability margins, the false-selection probabilities for noisy nodes in each view decay exponentially with respect to the gap between the clean and noisy loss means.

**(2) Dual-consistency contraction.** For the intersection event $A_g \cap A_s$, we have

$$\Pr(A_g \cap A_s) = \Pr(A_g)\Pr(A_s) + \mathrm{Cov}(\mathbf{1}_{A_g}, \mathbf{1}_{A_s}) \tag{15}$$

Since the two views rely on distinct information sources—global model prediction and structural propagation—their correlation over noisy nodes satisfies $\mathrm{corr}(A_g, A_s) < 1$. By the Fréchet–Hoeffding bound, $\Pr(A_g \cap A_s) \le \min\{\Pr(A_g), \Pr(A_s)\}$. If we further assume approximate conditional independence, the covariance term vanishes, yielding

$$\Pr(A_g \cap A_s) \approx \Pr(A_g)\Pr(A_s) \le \exp\left[-\frac{(\Delta_{m,c}^{(g)})^2}{2(\sigma_{m,c}^{n(g)})^2} - \frac{(\Delta_{m,c}^{(s)})^2}{2(\sigma_{m,c}^{n(s)})^2}\right] \tag{16}$$

Therefore, the dual-consistency mechanism achieves an *exponential contraction* of the false-selection probability compared with any single-view filtering. This theoretically supports the robustness of our proposed class-aware dual-consistency filtering under noisy and heterogeneous client conditions. ∎

---

**Algorithm 1** Training Algorithm of FedRGL

---

**Input:** Communication rounds $T_{\text{Max}}$, warm-up rounds $T_{\text{warm}}$, local epochs $E_{\text{Loc}}$, number of clients $M$, graph data with label noise $\{G_m\}_{m=1}^M$, local models $\{w_m\}_{m=1}^M$.
**Output:** Final global model $W^t$.

1: Warm up the global model using FedAvg for $T_{\text{warm}}$ rounds.
2: **for** $t = T_{\text{warm}} + 1$ **to** $T_{\text{Max}}$ **do**
3:     **Client Side:**
4:     **for all** $m = 1, 2, \ldots, M$ **in parallel do**
5:         Broadcast the global model $W^t$ to client $m$.
6:         Filter clean nodes $V_m^C$ and noisy nodes $V_m^N$ using Eqs. (1–6).
7:         **for** $e = 1, 2, \ldots, E_{\text{Loc}}$ **do**
8:             Select high-confidence nodes $V_m^P$ using Eq. (9).
9:             Constrain local training using Eq. (11).
10:        **end for**
11:        Compute the average predictive entropy $\mathcal{H}_m$ using Eq. (12).
12:     **end for**
13:     **Server Side:**
14:     Aggregate client models and update the global model $W^{t+1}$ using Eq. (13).
15: **end for**
16: **Return:** Final global model $W^{T_{\text{Max}}}$.

---

## E. Detailed Formulation of the Contrastive Loss

In this section, we provide the detailed formulation of the contrastive loss $L_{cl}$ used in our method. Following the GRACE framework (Zhu et al., 2020b), the contrastive loss is defined as:

$$L_{cl}\left(Z^1, Z^2\right) = -\sum_{i=1}^{N} \log \frac{\psi\left(Z_i^1, Z_i^2\right)}{\sum_{j=1, j \neq i}^{N} \psi\left(Z_i^1, Z_j^1\right) + \sum_{j=1}^{N} \psi\left(Z_i^1, Z_j^2\right)}, \tag{17}$$

where $\psi(a, b) = \exp(\mathbf{sim}(a, b)/\tau)$, $\mathbf{sim}(\cdot, \cdot)$ denotes the cosine similarity function, and $\tau$ is the temperature parameter, set to 0.5 in our experiments. This loss encourages the embeddings of the same node under dual augmented views to be close, while pushing apart those of different nodes, thereby enhancing the encoder's noise robustness.

## F. Detailed Formulation of the JS Divergence

The Jensen–Shannon (JS) divergence loss $L_m^{Js}$ for pseudo-labeled nodes $V_m^P$ is formulated as:

$$L_m^{Js}(\hat{P}_m, \hat{P}_m^1, \hat{P}_m^2) = \frac{1}{2}\left[\text{KL}(\hat{P}_m^1 \| \hat{P}_m) + \text{KL}(\hat{P}_m^2 \| \hat{P}_m)\right], \tag{18}$$

where $\hat{P}_m = \frac{1}{2}(\hat{P}_m^1 + \hat{P}_m^2)$ denotes the averaged class-probability distribution across two augmented graph views, and $\text{KL}(\cdot \| \cdot)$ is the Kullback–Leibler divergence. This objective enforces prediction consistency between the augmented views and the original graph, stabilizing pseudo-label learning during local training.

## G. Pseudo-code for FedRGL

To provide a clearer understanding of the FedRGL training process, we present the pseudo-code of FedRGL in Algorithm 1.

## H. Complexity analysis, privacy analysis, and limitations.

### H.1. Complexity Analysis

**Computational complexity analysis**. We analyze the complexity of the proposed FedRGL method. On the client side, the complexity of global model-based noisy node filtering is $O(N_{\text{train}}C)$, and for local substructures, it is $O(kN_{\text{train}}^2 C)$, where

*Table 5.* Total training time for 100 communication rounds (seconds)

| Dataset | FedAvg | FedTAD | FedCorr | CRGNN | RTGNN | ERASE | FedRGL |
|---------|--------|--------|---------|-------|-------|-------|--------|
| Cora | 7.1 | 364.1 | 10.8 | 27.2 | 19.1 | 51.2 | 45.5 |
| CS | 13.8 | >3600 | 30.7 | 73.6 | 37.3 | 516.9 | 225.7 |

*Table 6.* Training time and accuracy for different datasets under uniform and pair noise at different filter frequencies.

| Dataset | Filter Frequency | Training Time (s) | Accuracy (%) |
|---------|-----------------|-------------------|--------------|
| Cora Uniform | Every | 47.2 | 78.75 |
| | Ten | 29.9 ↓ **17.3** | 78.24 ↓ **0.51** |
| Cora Pair | Every | 45.5 | 75.14 |
| | Ten | 29.2 ↓ **16.3** | 74.27 ↓ **0.87** |
| CS Uniform | Every | 225.7 | 81.01 |
| | Ten | 81.2 ↓ **144.5** | 78.22 ↓ **2.79** |
| CS Pair | Every | 227.4 | 87.79 |
| | Ten | 83.6 ↓ **143.8** | 86.90 ↓ **0.89** |

$C$ is the number of classes and $k$ is the number of label propagation steps. The complexity of pseudo-label prediction and JS divergence is $O(2N_{\text{noisy}}C)$, while local prediction entropy is $O(N_{\text{unlabel}}C)$, where $N_{\text{noisy}}$ and $N_{\text{unlabel}}$ represent the numbers of noisy and unlabeled nodes, respectively. On the server side, aggregating parameters from $M$ clients has a complexity of $O(MP)$, where $P$ is the number of model parameters.

**Experimental validation of complexity**. We recorded the total training time over 100 communication rounds for various methods on Cora (5 clients) and CS (10 clients) with a pair noise rate of 0.3. As shown in Tab. 5, FedRGL slightly increases training complexity but achieves significant test accuracy improvements under label noise or maintains performance with clean labels (Tab. 2). To further optimize efficiency, we reduced the frequency of noise node filtering to once every **ten** communication rounds. Results in Tab. 6 demonstrate that this approach significantly reduces training time while maintaining nearly the same global model performance, proving the effectiveness of the proposed class-aware dynamic dual-view filtering. ★ *The implementation of reduced-frequency noisy node filtering is available through the code link provided in the abstract.*

**Communication Complexity**. Let $|\theta|$ denote the number of model parameters and $M$ be the number of active clients per round. In each communication round of FedRGL, in addition to the standard model parameters, each client only uploads a single scalar representing the average prediction entropy. Consequently, the communication overhead per client per round is $O(|\theta| + 1)$, which is asymptotically equivalent to the vanilla FedAvg, i.e., $O(|\theta|)$. This marginal $O(1)$ increase is practically negligible, ensuring that FedRGL is highly scalable and efficient for large-scale federated learning tasks.

### H.2. Privacy Analysis

FedRGL follows the standard federated learning protocol where clients upload local model parameters $\theta$. Beyond this, the only additional information shared is a single aggregated scalar: the average prediction entropy. Since this entropy value is a global summary statistic of the local model's uncertainty, it does not expose individual sample features or the local graph topology. Compared to methods that require sharing class-wise distributions or local prototypes, FedRGL significantly limits the auxiliary information leakage. This ensures that the privacy risk remains on par with standard FedAvg while providing the necessary information for noise-robust aggregation.

### H.3. Limitation

FedRGL proposes dual-perspective consistent noisy node filtering based on local substructures and global model information, assuming that the local client structures are not excessively perturbed. However, in more challenging real-world scenarios, such as when local subgraph structures are perturbed by adversarial attacks (*i.e.*, Mettack (Zügner & Günnemann, 2019)), the performance of FedRGL may suffer. This represents a dual-noise research scenario that warrants further exploration. Moreover, FedRGL does not explicitly address backdoor attacks (Dai et al., 2023; Xu et al., 2025), where noise contamination may be associated with specific trigger patterns. Enhancing its defense capability against such trigger-based attacks is also an important direction for future work.

*Table 7.* Statistics of the experimental datasets used in our study.

| Dataset | Nodes | Edges | Classes | Features | Type |
|---|---|---|---|---|---|
| Cora | 2,708 | 5,429 | 7 | 1,433 | Citation |
| CiteSeer | 3,327 | 4,732 | 6 | 3,703 | Citation |
| PubMed | 19,717 | 44,338 | 3 | 500 | Citation |
| CS | 18,333 | 81,894 | 15 | 6,805 | Co-author |
| Physics | 34,493 | 247,962 | 5 | 8,415 | Co-author |
| Photo | 7,487 | 119,043 | 8 | 745 | User-item |
| ogbn-arxiv | 169,343 | 2,315,598 | 40 | 128 | OGB (citation) |
| ogbn-products | 2,449,029 | 61,859,140 | 47 | 100 | OGB (product) |

## I. Datasets

The statistical details of the experimental datasets are summarized in Table 7, with additional descriptions provided below:

- **Cora:** A classic citation network where nodes represent scientific papers and edges denote citation relationships. It consists of 2,708 nodes and 5,429 edges. The nodes are categorized into 7 classes, with features represented as 1,433-dimensional vectors.

- **CiteSeer:** A citation network similar to Cora, containing 3,327 nodes and 4,732 edges. It has 6 categories, and each node is characterized by a 3,703-dimensional feature vector.

- **PubMed:** A large-scale biomedical citation network comprising 19,717 nodes and 44,338 edges. The nodes are divided into 3 classes with 500-dimensional features.

- **CS:** A co-authorship network in the computer science domain. Nodes represent authors, and edges indicate collaborations. It contains 18,333 nodes and 81,894 edges across 15 classes, with a feature dimension of 6,805.

- **Physics:** A co-authorship network in the physics domain, consisting of 34,493 authors (nodes) and 247,962 collaborations (edges). It features 5 classes and high-dimensional node features of 8,415.

- **Photo:** An Amazon co-purchase network where nodes are products and edges represent frequent co-purchases. It includes 7,487 nodes and 119,043 edges across 8 categories, with 745-dimensional features.

- **ogbn-arxiv:** A massive citation network from the Open Graph Benchmark (OGB), containing 169,343 nodes (papers) and 2,315,598 edges (citations). It is categorized into 40 subject areas with 128-dimensional embeddings.

- **ogbn-products:** The largest dataset in our study, representing an Amazon product co-purchase network from OGB. It comprises 2,449,029 nodes and 61,859,140 edges across 47 product categories.

## J. Noise Settings

In this paper, we consider three types of label noise: **Uniform**, **Pair**, and **Instance-dependent**. These noise types are widely adopted in studies on learning with noisy labels (Xu et al., 2022; Li et al., 2024c; Qian et al., 2023). The transition probability matrices for the first two are defined as follows:

①  Uniform Noise

$$
\mathbf{T}_{\text{uniform}} = \begin{bmatrix} 1-\eta & \frac{\eta}{C-1} & \cdots & \frac{\eta}{C-1} \\ \frac{\eta}{C-1} & 1-\eta & \cdots & \frac{\eta}{C-1} \\ \vdots & \vdots & \ddots & \vdots \\ \frac{\eta}{C-1} & \frac{\eta}{C-1} & \cdots & 1-\eta \end{bmatrix} \tag{19}
$$

②  Pair Noise

$$\mathbf{T}_{\text{pair}} = \begin{bmatrix} 1-\eta & \eta & 0 & \cdots & 0 \\ 0 & 1-\eta & \eta & \cdots & 0 \\ \vdots & \vdots & \vdots & \ddots & \vdots \\ \eta & 0 & 0 & \cdots & 1-\eta \end{bmatrix} \tag{20}$$

**Uniform noise** implies that a label from any given class is flipped uniformly at random with a probability of $\frac{\eta}{C-1}$ to any other class. **Pair noise** refers to a case where a label from one class is flipped to a specific other class with probability $\eta$. ③ **Instance-dependent noise** (*Instance Noise*) simulates more realistic scenarios where the probability of label corruption depends on instance-level features. For each node $v_i$, its true label $y_i$ is flipped to another label $y_j$ according to a softmax-normalized probability proportional to the cosine similarity between their feature representations.

**Noise rate sampling.** To simulate heterogeneous noise across clients, the noise rate for each client is uniformly sampled from the interval $U(\eta^l, \eta^u)$, where $\eta^l$ and $\eta^u$ denote the lower and upper bounds, respectively. In this study, the upper bound $\eta^u$ is set to 0.6 for Uniform noise, 0.45 for Pair noise, and 0.5 for Instance-dependent noise, while the lower bound equals the predefined noise rate $\eta$. ***Notably***, due to the structural heterogeneity among local subgraphs ($A_m \neq A_n$), even under identical noise rates, the specific noise patterns across clients can still differ significantly.

## K. Baseline Methods

We compare FedRGL with 12 baselines. The specific details of these baselines are as follows.

- **FedAvg (McMahan et al., 2017)**: Clients send all learnable parameters to the server and receive aggregated parameters from the server for the next round of training.

- **FGSSL (Huang et al., 2023)**: This method mitigates the impact of graph semantic and structural biases by introducing supervised graph contrastive learning and relational distillation during local training.

- **FedTAD (Zhu et al., 2024)**: FedTAD is a topology-aware data-free knowledge distillation framework for subgraph federated learning, which addresses subgraph heterogeneity by enhancing reliable class-wise knowledge transfer from local GNNs to the global model.

- **FedSPA (Tan et al., 2025b)**: FedSPA is a federated graph learning framework designed to address homophily heterogeneity across clients. It mitigates homophily conflict via Subgraph Feature Propagation Decoupling (SFPD) to align inconsistent homophily levels, and further reduces homophily bias through Homophily Bias-Driven Aggregation (HBDA), which adaptively adjusts client contributions during aggregation.

- **FedCorr (Xu et al., 2022)**: This method proposes a three-stage label noise correction approach, relying on the GMM algorithm to filter out noisy clients and noisy samples, and uses the global model to relabel the noisy samples. We adapted the publicly available code of this method to the subgraph federated learning scenario and performed the corresponding tuning.

- **FedNoro (Wu et al., 2023)**: This method handles label noise using distillation loss from the global model and reweighting based on parameter distance on the server. We modified the publicly available code for this method to apply to the subgraph federated learning scenario and performed the necessary tuning.

- **FedNed (Lu et al., 2024)**: This method requires additional unlabeled public datasets on the server side to perform negative distillation learning, preventing the global model from being influenced by high-noise clients' model parameters. To adapt this method to subgraph federated learning, we referred to the random subgraph construction method in FedPub (Baek et al., 2023) and constructed a random graph dataset on the server side to replace the original required labeled public dataset.

- **RHFL (Fang & Ye, 2022)**: By introducing a robust symmetric loss function during local training, this method regularizes the local model parameters and uses an unlabeled public dataset on the server for weighted model updates. Similarly, we used the data construction method from FedNed to adapt RHFL to the subgraph federated learning scenario.

- **CRGNN (Li et al., 2024c)**: In centralized graph learning with noisy labels, this method dynamically selects nodes with consistent predictions using graph contrastive learning to update the model and enhances the encoder with neighborhood contrast and cross-space consistency regularization. We implemented this method under the FedAvg framework.

- **RTGNN (Qian et al., 2023)**: This method, in centralized graph learning with noisy labels, enhances the graph structure, filters noisy nodes, and applies consistency regularization based on the characteristics of dual-network training. We implemented this method under the FedAvg framework.

- **ERASE (Chen et al., 2024)**: This method proposes a decoupled label propagation algorithm and a fused prototype pseudo-labeling approach to learn robust node representations (focusing mainly on training the encoder) in centralized graph learning with noisy labels. Since this method is not end-to-end training, we adapted it under the FedAvg framework to perform joint training of the encoder and classifier on each client to suit the federated learning scenario.

- **DND-NET (Ding et al., 2024)**: This method, originally designed for centralized graph learning with noisy labels, builds a noise-resilient architecture by eliminating the message-passing mechanism and introducing reliable pseudo labeling. To extend its robustness to federated settings, we adapt the method under the FedAvg framework, where each client performs local training using the denoising backbone and local pseudo label refinement. This allows for effective collaborative training while mitigating the propagation of local label noise across clients.

## L. Detailed Implementation and Hyperparameter Settings

All clients are trained using SGD with a learning rate of $1 \times 10^{-2}$ and a weight decay of $5 \times 10^{-4}$. The total number of communication rounds is set to 100, with 3 local training epochs per round. Hyperparameters of FedRGL are tuned using the Optuna framework (Akiba et al., 2019). Specifically, $\varphi_1$ and $\varphi_2$ are selected from $\{0.5, 1.0, 1.5, 2.0\}$. The high-confidence pseudo-label threshold $\gamma$ is searched within $\{0.5, 0.6, 0.7, 0.8, 0.9, 0.95\}$. The parameter $k$ is explored in $\{5, 8, 10, 15\}$. In addition, the trade-off parameters $\lambda_{\text{CL}}$, $\lambda_p$, and $\lambda_{\text{Js}}$ are tuned in the range $\{0.1, 0.2, 0.3, \ldots, 1.0\}$. All experiments are conducted on an NVIDIA A100-PCIE-40GB GPU. ★ The optimal hyperparameter configurations for each dataset, together with the complete search space, are provided in the released code for reproducibility.

*Table 8.* Performance comparison of different methods under instance-dependent label noise.

| Method | Cora | CS | Physics | Ogbn-products |
|---|---|---|---|---|
| FedAvg | $47.39 \pm 0.52$ | $65.03 \pm 0.42$ | $67.39 \pm 0.71$ | $57.35 \pm 0.28$ |
| FedCorr | $51.21 \pm 0.44$ | $69.79 \pm 0.92$ | $74.75 \pm 0.55$ | $58.21 \pm 0.45$ |
| FedNed | $47.85 \pm 0.37$ | $66.51 \pm 0.39$ | $68.77 \pm 0.62$ | $57.73 \pm 0.52$ |
| CRGNN | $60.76 \pm 0.72$ | $82.11 \pm 0.46$ | $79.63 \pm 0.61$ | $59.58 \pm 0.71$ |
| DND-NET | $57.25 \pm 0.66$ | $72.25 \pm 0.93$ | $77.28 \pm 1.11$ | $59.77 \pm 0.66$ |
| **FedRGL** | $\mathbf{76.29 \pm 0.54}$ | $\mathbf{86.27 \pm 0.55}$ | $\mathbf{85.21 \pm 0.63}$ | $\mathbf{63.45 \pm 0.57}$ |

## M. Instance-dependent Noise Results

We further evaluate the performance of FedRGL under instance-dependent label noise with a noise rate of 0.3, as shown in Tab. 8. Compared with the baselines, FedRGL achieves the best results across all datasets, including the large-scale *ogbn-products*. While traditional methods such as FedCorr and DND-NET show partial improvements, they fail to handle complex instance-dependent corruption effectively. In contrast, FedRGL leverages dual-consistency filtering and class-aware dynamic thresholds to maintain stable accuracy, surpassing **FedAvg** by more than 6% on average and demonstrating strong robustness against feature-related noise perturbations.

## N. Validation under Different Noisy Client Ratios and Participation Rates

**Different Noisy Client Ratios.** To evaluate the performance of FedRGL with varying proportions of noisy clients, we conducted experiments on the Photo dataset with 20 clients (Pair noise type, noise rate $\eta = 0.4$). The experimental results are presented in Tab. 9, where $x\%$ indicates the proportion of noisy clients. From the results in the figure, it can be observed that FedRGL achieves the best prediction accuracy across all noisy client proportions. Furthermore, as the proportion of noisy clients increases, the performance gap between FedRGL and other baseline methods widens, further demonstrating

*Table 9.* Accuracy on Photo with different numbers of noisy clients.

| Method | 20% | 40% | 60% | 80% | 100% |
|---|---|---|---|---|---|
| FedAvg | $85.7 \pm 0.1$ | $80.9 \pm 1.0$ | $76.6 \pm 0.2$ | $72.0 \pm 1.5$ | $53.4 \pm 1.7$ |
| FedCorr | $78.0 \pm 1.3$ | $75.0 \pm 2.8$ | $61.4 \pm 1.4$ | $61.3 \pm 2.2$ | $48.9 \pm 3.4$ |
| FedNoro | $61.9 \pm 0.4$ | $56.3 \pm 1.3$ | $48.7 \pm 1.4$ | $47.7 \pm 0.7$ | $39.8 \pm 0.3$ |
| CRGNN | $85.5 \pm 0.5$ | $78.6 \pm 2.0$ | $72.3 \pm 1.5$ | $73.9 \pm 0.8$ | $58.8 \pm 1.6$ |
| DND-NET | $84.3 \pm 0.7$ | $74.3 \pm 0.4$ | $73.2 \pm 0.3$ | $69.8 \pm 0.3$ | $60.3 \pm 0.5$ |
| **FedRGL** | $87.5 \pm 0.1$ | $87.4 \pm 0.2$ | $82.3 \pm 0.3$ | $79.1 \pm 0.1$ | $73.4 \pm 0.6$ |

*Table 10.* Accuracy under Different Client Participation Rates on Physics.

| Dataset | Physics (20 Clients) | | | |
|---|---|---|---|---|
| **Method** | 40% | 60% | 80% | 100% |
| FedAvg | $61.58 \pm 0.52$ | $62.93 \pm 0.38$ | $63.08 \pm 0.49$ | $65.27 \pm 0.43$ |
| FedCorr | $63.72 \pm 0.89$ | $65.31 \pm 0.96$ | $67.55 \pm 0.82$ | $71.42 \pm 0.77$ |
| CRGNN | $68.81 \pm 0.77$ | $72.90 \pm 0.55$ | $74.36 \pm 0.41$ | $77.25 \pm 0.37$ |
| DND-NET | $64.52 \pm 1.21$ | $68.73 \pm 1.07$ | $71.29 \pm 0.96$ | $75.09 \pm 1.25$ |
| **FedRGL** | $\mathbf{82.11 \pm 0.62}$ | $\mathbf{85.26 \pm 0.64}$ | $\mathbf{87.37 \pm 0.79}$ | $\mathbf{88.76 \pm 0.63}$ |

the effectiveness of the proposed method in identifying and correcting noisy nodes. Additionally, it is worth noting that compared to methods requiring additional steps to filter clean and noisy clients (*e.g.,* FedCorr and FedNoro), FedRGL omits this step. Through dual-view consistency filtering combined with the class-aware filtering threshold, FedRGL can directly and accurately identify noisy nodes. Moreover, it effectively utilizes the supervision information from clean clients to train local models without compromising their model performance, as further confirmed by the performance of FedRGL under the Normal setting in Tab. 2.

**Participation Rates.** In federated learning, training with a subset of sampled clients in each communication round is an effective strategy to reduce communication overhead. To evaluate the prediction performance of FedRGL under this setting, we conducted experiments on the Physics dataset with 20 clients (Uniform noise type, noise rate = 0.5) using four different client participation rates. The results are presented in Tab. 10. From the results, it can be seen that FedRGL achieves the best prediction accuracy across all participation rates, demonstrating strong robustness and stability. Moreover, all methods exhibit improved test accuracy of the global model as the client participation rate increases.

## O. Diagnostic Analysis

### O.1. Ablation Experiment

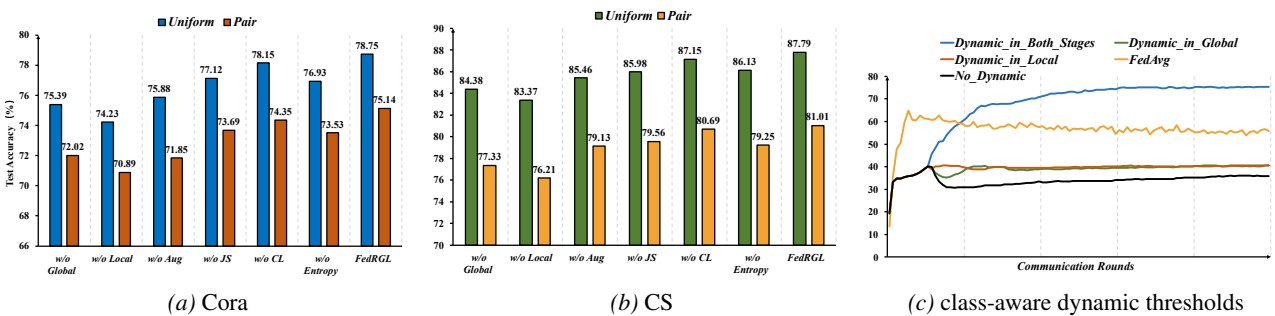

*Figure 8.* (a) and (b): Ablation Study of FedRGL on the Cora and CS Datasets. (c): Validation of Class-Aware Dynamic Filtering Threshold.

We conduct ablation studies on the key components of FedRGL using the Cora dataset with 5 clients and the CS dataset with 10 clients under a noise rate $\eta = 0.3$. Specifically, we evaluate six FedRGL variants, as shown in Fig. 8. ❶ *w/o Global*:

Removing the global model perspective for noisy-node filtering causes a notable accuracy drop, highlighting the importance of leveraging global model knowledge for noise identification. ❷ *w/o Local*: Removing the local substructure perspective leads to a more significant performance degradation, especially under pair noise, demonstrating that local graph information plays a critical role in identifying noisy nodes in subgraph FL. ❸ *w/o Aug*: Removing the graph augmentation branch disables dual-view predictions for high-confidence pseudo-label supervision, so the model mainly relies on the filtered clean-label nodes for local training. The performance drop of this variant confirms that graph augmentation is important for recovering reliable supervision from selected noisy nodes. ❹ *w/o JS*: Excluding the JS divergence constraint degrades performance, indicating that two-view prediction consistency helps stabilize the supervision of high-confidence noisy nodes. ❺ *w/o CL*: Removing the graph contrastive learning loss causes only minor performance fluctuations compared with removing the graph augmentation branch. This suggests that contrastive learning is helpful for representation regularization, but it is not the dominant source of improvement. ❻ *w/o Entropy*: Removing prediction-entropy-based robust aggregation lowers the global model quality, showing its role in reducing the influence of uncertain client updates under subgraph heterogeneity.

To further validate the effectiveness of the class-aware dynamic loss threshold in dual-view noisy-node filtering, we test four settings on the Cora dataset with 5 clients under pair noise at $\eta = 0.3$: 1) *Dynamic in Both Stages*, where both global and local perspectives use dynamic thresholds; 2) *Dynamic in Global*, where only the global perspective uses dynamic thresholds; 3) *Dynamic in Local*, where only the local perspective uses dynamic thresholds; and 4) *No Dynamic*, where conventional GMM-based filtering is used. The results in Fig. 8c show that the proposed class-aware dynamic thresholds are crucial for performance improvement. Static GMM-based filtering fails to bring clear gains and performs much worse than FedRGL. Moreover, applying dynamic thresholds to only one perspective is insufficient, since the final clean-node set is obtained through the intersection of global and local filtering results.

## O.2. Parameter Sensitivity Analysis

In this section, we explore the sensitivity of the model to hyperparameters. Specifically, we focus on investigating the class-aware dynamic filtering thresholds $\varphi_1$ and $\varphi_2$, the number of label propagation steps $k$ in the local substructure, the filtering threshold $\gamma$ for high-confidence noisy nodes, and the supervision loss hyperparameter $\lambda_P$ for pseudo-labeled nodes. For the contrastive loss hyperparameter $\lambda_{Cl}$ and the JS divergence loss hyperparameter $\lambda_{Js}$, the former has a negligible impact on the model and is thus not discussed here. The latter was tuned using Optuna (Akiba et al., 2019), and the optimal value was found to be either 0.3 or 0.4 across all datasets. *The optimal values for all hyperparameters in the datasets can be found in the provided code*.

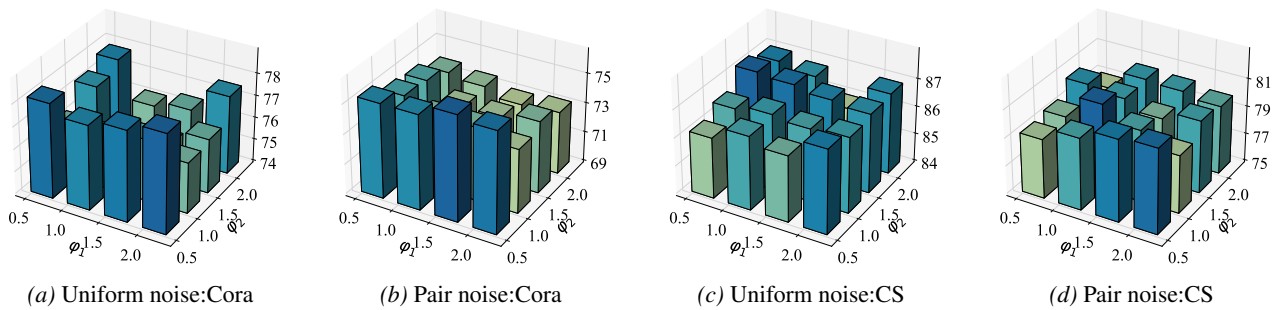

*(a)* Uniform noise:Cora     *(b)* Pair noise:Cora     *(c)* Uniform noise:CS     *(d)* Pair noise:CS

*Figure 9.* The performance impact of class-aware dynamic filtering thresholds $\varphi_1$ and $\varphi_2$ on the model is validated on the Cora and CS datasets under two noise types with a noise rate of 0.3.

Fig. 9 shows the test accuracy of FedRGL under different combinations of class-aware dynamic filtering thresholds $\varphi_1$ and $\varphi_2$. The results indicate that FedRGL performs more stably on the CS dataset than on Cora, where fewer training nodes and incomplete graph structure across clients cause performance fluctuations. Importantly, under all combinations of $\varphi_1$ and $\varphi_2$, FedRGL consistently outperforms the 12 baseline methods (Tab. 2). For instance, on the Cora dataset with Uniform noise, FedRGL achieves an average accuracy of **75.27%** under the worst hyperparameter combination, significantly higher than the next-best CRGNN at **61.67%**. This highlights the effectiveness of the proposed bi-perspective consistency-based dynamic filtering method in improving global model performance for federated graph node classification tasks with noisy labels. ***Notably***, even simplifying the hyperparameter settings to $\varphi_1 = \varphi_2$ still yields competitive accuracy.

Figs. 10a and 10b illustrate the impact of label propagation iterations on FedRGL across the Cora, CS, and Photo datasets.

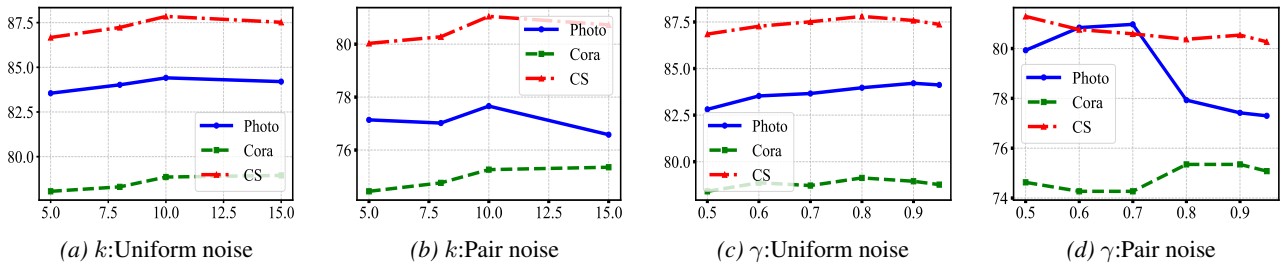

*(a) $k$:Uniform noise*    *(b) $k$:Pair noise*    *(c) $\gamma$:Uniform noise*    *(d) $\gamma$:Pair noise*

*Figure 10.* (a) and (b) show the impact of the number of label propagation iterations $k$ on the Cora, CS, and Photo datasets under Uniform and Pair noise types with a noise rate of 0.3 for the FedRGL method. (c) and (d) illustrate the impact of high-confidence pseudo-label prediction $\gamma$ on the Cora, CS, and Photo datasets under Uniform and Pair noise types with a noise rate of 0.3 for the FedRGL method.

The results show that FedRGL is largely insensitive to the hyperparameter $k$. After 5 iterations, the model achieves strong robustness and high predictive performance. While increasing iterations further improves results on Cora and CS, excessive iterations on Photo may slightly degrade performance. Figs. 10c and 10d show the effect of high-confidence pseudo-label filtering thresholds $\gamma$. FedRGL's performance remains stable on Cora and CS, but on Photo under Pair noise, performance gradually declines as $\gamma$ increases. Nonetheless, even at its lowest performance, FedRGL still significantly outperforms the 12 baseline methods, indirectly demonstrating the stability of the proposed method.

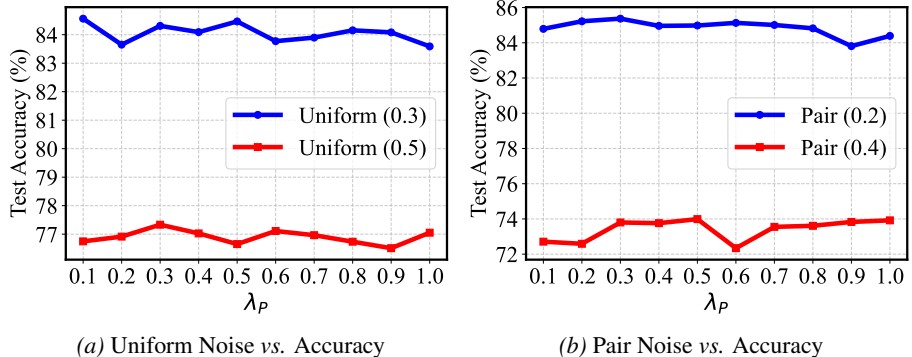

*(a)* Uniform Noise *vs.* Accuracy    *(b)* Pair Noise *vs.* Accuracy

*Figure 11.* To validate the impact of the pseudo-labeled node supervision loss hyperparameter $\lambda_P$ on FedRGL across the Photo dataset under different noise types and noise rates.

We further validated the impact of the high-confidence pseudo-labeled node supervision loss hyperparameter $\lambda_P$ on FedRGL, as shown in Fig. 11. The results indicate that the hyperparameter $a$ introduces certain performance fluctuations for FedRGL under different noise types and noise rates. However, these fluctuations remain within an acceptable range. *Overall*, the proposed FedRGL algorithm effectively enhances the prediction performance of the federated global model in noisy label environments, thanks to its key components, advancing the robustness research in federated graph learning.

