# OpenReview forum: "FedRGL: Robust Federated Graph Learning under Label Noise"
_ICML.cc/2026/Conference — ICML 2026 regular_

### Official Review · Reviewer_gnho · 2026-03-08

**Soundness:** 4
**Presentation:** 4
**Significance:** 4
**Originality:** 3
**Overall Recommendation:** 5
**Confidence:** 5

**Summary:**

This paper proposes an innovative method (FedRGL) to enhance the robustness of subgraph federated learning under noisy labels. The method introduces a dual-view noise node filtering strategy that integrates both the global model perspective and the local subgraph structural perspective, together with a class-wise dynamic threshold to improve the accuracy of noise detection. Furthermore, this filtering module can be used as a plug-and-play component to strengthen the noise robustness of existing subgraph federated learning frameworks. In addition, a graph contrastive learning–based enhancement is employed to obtain high-quality pseudo-labels for the filtered noisy nodes. Finally, the average prediction entropy of unlabeled nodes is utilized to evaluate the reliability of client models under noisy conditions, enabling robust global aggregation. Extensive experiments on eight graph datasets, including two large-scale OGBN benchmarks and twelve representative baseline methods, across various client numbers, noise rates, noise types, and noisy-client ratios, demonstrate that the proposed method achieves excellent performance, stability, and scalability in improving the noise-label robustness of subgraph federated learning.

**Compliance With Llm Reviewing Policy:**

Affirmed.

**Final Justification:**

This paper studies federated graph node classification under noisy labels and proposes FedRGL to improve robustness in the noisy subgraph FL setting. I find the paper practically meaningful, and the overall method design is technically sound. The experimental results are also fairly strong, showing consistent advantages across multiple datasets and settings. Beyond the performance gains of the full method itself, CADF, as the core design, also demonstrates good plug-and-play capability, as it can be effectively integrated into existing subgraph FL frameworks and still bring stable improvements, which further strengthens the practical value of the work.

My initial concerns mainly focused on the overhead issue on large-scale graphs, the generalizability under different graph partition strategies, the explanation of the results under the Normal setting, the computational complexity analysis, and the reasonableness of using validation nodes to evaluate local training quality for server-side weighting. After reading the authors’ response, I believe these concerns have been addressed to a sufficient extent. In particular, the additional analysis on large-scale graphs, the supplementary experiments under Metis partition, and the clarification of the relevant code locations have significantly improved my confidence in the correctness of the implementation, as well as the scalability and generalizability of the method. The explanations regarding the Normal setting, the complexity analysis, and the reasonableness of validation-node-based weighting also made the technical details much clearer to me.

Overall, the authors’ response did not change my originally positive assessment, but it further reinforced it. Although some of these clarifications should still be incorporated into the revised version to improve the paper’s clarity, my main concerns have been resolved. Therefore, my final position is supportive of acceptance.

**Key Questions For Authors:**

1. What explains the strong generalization ability of FedRGL under the Normal setting in Table 2, where the training nodes are free from noisy labels?
2. Why can the information from local validation nodes be reliably used to evaluate the training quality of local models at each client? In addition, are the computed entropy values themselves susceptible to noise?
3. Graph contrastive learning is often computationally expensive and may even cause OOM issues on large-scale datasets such as the OGBN benchmarks used in this paper. How does the proposed method address or alleviate this issue in practice?

**Limitations:**

Yes

**Strengths And Weaknesses:**

**Strengths:**
1、This work investigates noisy-label learning in federated graph learning, a setting that is fundamentally different from federated learning in the vision domain and centralized graph node classification. Figs. 1 and 2 clearly present the motivation, key challenges, and the limitations of existing methods. The paper is generally well organized, with a coherent methodological design and clear presentation.

2、The proposed FedRGL method introduces a dual-end consistency-based noisy node screening module(CADF), which can effectively identify and filter noisy nodes under the dual challenges of incomplete subgraph structures and noisy training labels. The noisy node screening results in Table 1 further demonstrate the advantage of dual-end consistency for noise localization. In addition, the method incorporates graph contrastive learning for augmentation and assigns high-confidence pseudo-labels to the filtered nodes, which is a reasonable and creative design choice, reflecting the distinctiveness and innovation of the proposed method compared with existing approaches.

3、The CADF module in the proposed FedRGL method is plug-and-play and can be readily integrated into seven existing federated graph learning methods (FGSSL, FedTAD, FedSPA, FedSage+, FedLoG, FedGTA, and FedATH). It improves the robustness of their global models against noisy labels and achieves accuracy gains of 16.65%–19.55%, demonstrating the strong scalability and robustness of the proposed method.

4、The paper provides theoretical analysis to justify the noise-screening capability of CADF and derives an exponential upper bound on the error shrinkage under the dual-end view, thereby supporting the effectiveness of the proposed method from both the research motivation and theoretical perspective. More importantly, the authors provide reproducible code and report the optimal hyperparameter settings on different datasets.

5、The experiments are comprehensive. The paper compares the proposed method with 12 strong baselines on eight graph node classification datasets of different scales and characteristics. It also includes extensive evaluations under varying numbers of clients, noise rates, numbers of noisy clients, and client online ratios, showing that FedRGL achieves strong performance under diverse heterogeneity and noise settings. In addition, the ablation results in Fig. 6 indicate that the proposed method has good convergence behavior and stable performance with respect to hyperparameter changes.

**Weaknesses:**
1、The experimental section adopts the commonly used Louvain algorithm to partition client subgraphs, but lacks validation under other graph partitioning strategies, such as Metis. It would strengthen the paper to include additional experiments under different partitioning methods to further verify the robustness and generalizability of the proposed method.

2、In Section 4.2, the analysis of the main experimental results in Table 2 is relatively limited. In particular, the paper provides insufficient quantitative discussion of the performance differences and lacks a deeper explanation of why the proposed method achieves these improvements. If space is limited, the authors are encouraged to provide a more detailed analysis in the appendix.

3、Although Appendix H.1 provides the client-side computational complexity and server-side communication overhead of FedRGL, the derivation of these complexity results is not sufficiently explained. For example, for the local substructures term $O(kN_{\text{train}}^{2}C)$, the paper does not clearly show how this complexity is derived, and the definition of $C$ is also missing. The computational complexity analysis should be further clarified and strengthened for better rigor and readability.

---

> ### Author Rebuttal · Authors · 2026-03-31
>
> **Dear Reviewer gnho,**
>
> We thank you for your careful review and your recognition of our work. We will address your questions point by point.
>
> **W1: Metis split validation**
>
> **A1:** Thank you for the suggestion. Both Louvain and Metis partitioning can allocate client data for federated graph node classification [1], with Louvain more commonly used. To validate FedRGL under Metis splits, we kept the same settings as Tab.2. See Tab. A in the anonymous link for detailed results: https://anonymous.4open.science/r/Tab-D-EE4E/README.md. Results show FedRGL maintains robust performance and generalization under Metis partitioning.
>
> **W2&Q1: Table 2 Performance Analysis**
>
> **A2:** Under clean-label settings, FedRGL performs well because client subgraphs still have structural incompleteness and class imbalance, causing nodes to learn at different rates (Fig.1). CADF dual-view selection, high-confidence pseudo-labeling for noisy nodes, and predictive-entropy-based aggregation across clients focus training on confident nodes while mitigating uncertain ones, enhancing overall global model performance. A detailed analysis of Tab.2 will be provided in a future revision.
>
> **W3: Complexity Analysis**
>
> **A3:** Thank you for your comment. The local substructure complexity formula in App.H.1 is derived from the client performing k steps of label propagation on its local subgraph, with each step requiring class-aware aggregation over all training nodes. Each client has *N_train* training nodes and *C* classes, resulting in a total computational complexity of *O(k N_train² C)*, reflecting the effect of subgraph size, class number, and propagation steps on local computation. We will clarify the formula and parameter definitions in a future revision.
>
> **Q2: Validation reliability**
>
> **A4:** FedRGL and most federated graph node classification methods [1] adopt transductive training. In this setting, all node features and graph structure are visible during training, but only training node labels are used for supervision. Therefore, the features and structure of local validation nodes can be used to compute predictive entropy for evaluating model quality without accessing their labels. Predictive entropy is used only for server-side aggregation weighting, and averaging across clients smooths minor noise, ensuring global model stability.
>
> **Q3: Graph contrastive learning on large-scale graphs**
>
> **A5:** Thank you for this issue. In FedRGL, graph contrastive learning is used after CADF completes noisy node filtering to generate two-view predictions for assigning high-confidence pseudo-labels and to enhance the encoder's robustness on standard-scale datasets. We also acknowledge that directly introducing contrastive loss on large-scale graphs could indeed incur additional computation and memory overhead.
>
> Regarding computation and memory concerns on large graphs, we clarify that FedRGL studies the **federated scenario**, and each client only holds a local subgraph obtained from global graph partitioning. Therefore, the number of nodes and edges participating in training on the client side is far **smaller than** the full graph. Even for large-scale graphs such as ogbn-arxiv and ogbn-products, graph augmentations, two-view predictions, and pseudo-label assignments are all performed on local subgraphs. Furthermore, in our two large-scale dataset experiments, **the contrastive loss term was not actually enabled for optimization**. For example, in the provided code (`train_FedRGL_arxiv.py`), the weight of the contrastive loss is set to 0.0 at line 61, and line 232 shows that the weight only affects the corresponding term in the total loss without impacting optimization. *We apologize that the paper did not explicitly describe this setting, and we will include the relevant description in subsequent updates.* We also acknowledge that during the code consolidation process, the intermediate variable `con_loss` at line 207 was not commented, which could lead to misunderstanding of the actual large-scale experiment configuration. In practice the corresponding loss weight is 0.0 and does not participate in the final optimization，this will be corrected in subsequent code releases to avoid ambiguity.
>
> We further provide experimental validation on large-scale graphs, including per-client average and maximum training time per epoch, as well as average and maximum memory usage. The experimental validation includes three configurations: single-view execution, dual-view without contrastive loss, and full configuration including the contrastive loss term. Due to space limits, the detailed results are available in Tab. B at https://anonymous.4open.science/r/Tab-D-EE4E/README.md. The results show that on large-scale datasets ogbn-arxiv and ogbn-products, FedRGL configured with dual-view without contrastive loss has computation and memory overheads that remain controllable.
>
> [1] OpenFGL: A Comprehensive Benchmark for Federated Graph Learning, VLDB'25

---

> > ### Author Rebuttal · Reviewer_gnho · 2026-04-03
> >
> > Thank you for taking the time to respond to the weaknesses and questions I raised in the previous stage.
> >
> > My main concerns have now been addressed. In particular, the additional memory analysis on large-scale graph datasets and the experimental validation under another graph partition method (Metis) have resolved my concerns regarding the scalability and generalizability of FedRGL, respectively. I also checked the code locations mentioned in your response, which further confirmed the correctness of the implementation on large-scale graphs. If space allows, I encourage you to incorporate these two points into the revised version to improve clarity and avoid possible misunderstanding.
> >
> > In addition, I now fully understand your explanation of the performance under the Normal setting in Table 2, as well as the computational complexity analysis in Appendix H.1. Please further supplement these parts in the revised version based on your current response.
> >
> > Finally, your clarification on using the predictive entropy of local validation nodes for server-side weighting has also addressed my concern about the training paradigm. Please make the applicable training setting of each method more explicit in the revised version, for example, whether it is limited to the transductive setting or is compatible with both transductive and inductive settings.
> >
> > Overall, I do not have further concerns about this paper. Your additional response has also reinforced my initial positive judgment of the paper. Therefore, I will keep my current score, which is already quite high, and increase my confidence in that score to indicate my support for accepting this paper.

---

> > > ### Author Response · Authors · 2026-04-06
> > >
> > > **Dear Reviewer gnho,**
> > >
> > > Thank you very much for your follow-up response and your strong recognition of our work. We are greatly encouraged to know that our clarifications and additional experiments have addressed your main concerns.
> > >
> > > We also sincerely thank you for your helpful suggestions for the revision. If we have the opportunity to revise the paper, we will further improve its clarity accordingly.

---

### Official Review · Reviewer_L2bW · 2026-03-12

**Soundness:** 3
**Presentation:** 3
**Significance:** 2
**Originality:** 2
**Overall Recommendation:** 4
**Confidence:** 5

**Summary:**

The paper tackles the label noise issue in Subgraph Federated Learning. The paper proposed to use Class-Aware Dual-consistency Filtering (CADF) module with both global (by model prediction) and local view (by Label propagation) to assign and correct the noisy labels. A graph contrastive learning approach is sequentially adopted to enhance the learning. For model aggregation, this paper uses the model prediction entropy on the test set to assign aggregation weights.

**Compliance With Llm Reviewing Policy:**

Affirmed.

**Final Justification:**

Concerns solved: inductive settings, Cost analysis

Concerns remain: limited innovation (CADF), additional complexity from three loss module.

Considering the efforts, I would like to improve my rating toward slightly positive. Overall, I think this is a borderline paper.

**Key Questions For Authors:**

1.	How to select the value of all hyperparameters ($\phi_1, \phi_2, \alpha, \gamma, \lambda_{CL}, \lambda_{P}, \lambda_{JS}$ )?

**Limitations:**

See Weakness 1

**Strengths And Weaknesses:**

Strength:

1.	This paper included analysis covering the noisy FL and noise labels in Graph learning methods.
2.	The experimental results include the large-scale graphs and different kinds of label noises (uniform, pair, and instance-dependent noise).
3.	The proposed method could be used as a plug-and-play module to enhance other FGL methods.

Weaknesses:

1.	The main issue is that the proposed method is only compatible with the transductive setting. Both Label propagation and model aggregation need additional unlabeled data.
2.	The graph contrastive learning module is not original and is always with high complexity regarding the computation and GPU memory footprint. This paper uses GRACE method (randomly mask edges and features). It constructs 2 augmented graph views, and 2 copies of embedding are needed. On smaller graphs this is ok. On ogbn-arxiv and ogbn-products, the additional costs are significant. Also, to use contrastive learning (Eq. 11), 3 additional losses are added, and 4 additional hyper-parameters are introduced.
3.	For label noise or scarcity, use label propagation, model prediction as soft labels. To enhance representation learning, use GRACE-based graph contrastive learning. To aggregate client models, use entropy-based aggregation. These are well-established methods. It’s understood that sometimes using existing simple, yet robust techniques is enough and practical. But to some extent it reflects this problem is not challenging and a good solution is to combine several proven solutions.

---

> ### Author Rebuttal · Authors · 2026-03-31
>
> **Dear Reviewer L2bW,**
>
> We appreciate your review and recognition of the significance of noisy subgraph federated learning, our large-scale and multi-noise evaluations, and the method’s plug-and-play potential. We address your main concerns below.
>
> **W1: Transductive setting and unlabeled nodes**
>
> **A1:** We appreciate your question. We acknowledge that FedRGL’s server-side entropy-based aggregation primarily suits transductive training (specifically explained in the note on p.2), which aligns with most existing centralized and federated graph node classification tasks based on the transductive training paradigm. However, the core CADF method of FedRGL is **not limited to transductive scenarios**: label propagation within local subgraphs does not depend on unlabeled nodes. Specifically, on p.4, right column, lines 194–200, we utilize subgraph masking to ensure that label propagation occurs only among training nodes, avoiding influence from non-training nodes (the corresponding implementation in code `util/task_util.py` lines 168–171 can also be reviewed).
>
> We further verified CADF + high-confidence pseudo-labels under inductive settings (other experimental settings same as Table 2). Due to space limits, results are in Table A at the anonymous link: https://anonymous.4open.science/r/Tab-C-4569/README.md. The results show that even without entropy aggregation, FedRGL still achieves significant performance gains across different noise types, demonstrating its effectiveness in both transductive and inductive scenarios for enhancing global model robustness against noisy labels.
>
> **W2: Contrastive cost and Eq.(11)**
>
> **A2:** We apologize for any confusion. Due to space limits, we respectfully refer you to our response to Reviewer gnho **Q3** for detailed discussion and experimental validation of the graph contrastive learning cost. Here, we focus on Eq.(11), which represents the full local training objective. Specifically, $\lambda_{cl} L_m^{CL}$ is the constraint from the graph contrastive loss, $\lambda_p L_m^P$ is high-confidence pseudo-label supervision, and $\lambda_{js} L_m^{JS}$ is two-view prediction consistency. **Only the first term is the constraint from graph contrastive loss**; the other two terms rely on predictions from graph-augmented views and can still be executed even if the contrastive loss constraint is removed, providing supervision for local training and stabilizing pseudo-label assignment.
>
> **W3: Novelty of the overall design**
>
> **A3:** The core innovation of FedRGL lies in the design of Class-Aware Dual-Consistency Filtering (CADF). Specifically, Fig.1 illustrates the challenges in noisy subgraph federated node classification: incomplete client subgraph structures, imbalanced class distributions, and coexisting noisy labels, with further details in App. C. These challenges make existing FNL and centralized GNL methods hard to directly transfer to federated graph scenarios, as validated by Fig.2.
>
> To address this, CADF combines global model prediction loss with local subgraph label propagation loss to extract clean node information from both global and local perspectives, and precisely identifies noisy nodes under class-aware dynamic thresholds. *The differences between FedRGL and existing GNL methods are directly discussed in p.3, right column, lines 151–153 and App. B*: 1) FedRGL does not rely on clean label priors; 2) initial labels are evaluated and corrected using a robust global model before propagation; 3) propagated soft labels, rather than one-hot labels, are used to compute the loss for noisy-node filtering. Thus, FedRGL is not a direct transfer of existing label propagation methods but modifies initialization and usage strategies.  Furthermore, CADF filters noisy nodes and, combined with graph contrastive augmentation, provides local supervision to extract clean signals from noisy nodes, enhancing local training. Client quality is measured via average prediction entropy on unlabeled nodes, using the transductive setting without extra computation or privacy cost, ensuring robust global aggregation.
>
> Finally, CADF is a hot-pluggable module for multiple existing subgraph FL frameworks, improving robustness under noisy labels，e.g., Tab.11 in App. O shows an average 16.65%–18.52% improvement across four frameworks. *FedRGL is not a simple combination of tools but a CADF-centered approach for noisy subgraph FL, providing a new technical route for federated graph learning.*
>
> **Q1: Hyperparameter selection**
>
> **A1:** Implementation Details (p.7) provides the basic experimental settings, and Appendix L lists the search ranges for all hyperparameters. We use the **automated hyperparameter optimization tool Optuna** to systematically search for the best parameters for each dataset, and the anonymous code includes these optimal parameters for reproducibility. Diagnostic Analysis (p.8) and App. Q.2 show that our method achieves stable and high performance across most parameter ranges.

---

> > ### Author Rebuttal · Reviewer_L2bW · 2026-04-02
> >
> > 1. On the large-scale graphs, why dual_view_no_cl incurs marginal additional overhead compared with single-view? Considering Eq.11, without cl, the forward, embedding, and 2 augmented views(large adjcency matrix and feature matrix) still exist. The additional cost is often not marginal. More decomposed analysis is required.
> > 2. As said, the core innovation is CADF. But CADF is still a framework considering both model-based cross-entropy (by model prediction) and self-structure-based cross-entropy (by LP). It's more like a refinement of a known solution.
> > 3. For inductive settings, it's suggested to evaluate on Flickr and Reddit datasets (they are used in evaluating many inductive models).

---

> > > ### Author Response · Authors · 2026-04-06
> > >
> > > **Dear Reviewer L2bW,**
> > >
> > > *Thank you for your response. We further clarify these points and provide additional evidence.*
> > >
> > > **W1: Computation cost**
> > >
> > > **A1:** We appreciate this helpful concern. We agree that, even without the contrastive loss term, **dual_view_no_cl** still involves two augmented views, intermediate node embeddings, and additional forward computation. Therefore, compared with **single-view**, its overhead is not negligible. To make this point clearer, we would like to further clarify that our setting is *federated subgraph learning*, rather than centralized whole-graph training, so the overhead here should be understood under Louvain-based client partition, rather than over the full *global graph*. For example, in the centralized setting, the ultra-large-scale graph dataset *ogbn-products* contains *2,449,029* nodes and *61,859,140* edges, while under our partition with *40 clients*, the largest client subgraph contains 61,251 nodes (client 38). and 4,377,362 edges (client 31). This suggests that the client-side subgraph is much smaller than the full graph.
> > >
> > > We would also like to clarify that, in the actual implementation, the graph structure (*edge*) is not stored or processed as an explicit dense **adjacency_matrix**, but in the sparse **edge_index** format. This can be seen in `train_FedRGL_arxiv.py`. At line 204, the local subgraph input is defined as `x, edge_index, y = subgraph.x, subgraph.edge_index, subgraph.y`, and line 206 calls `forward_two_views(x, edge_index)`. Therefore, the graph structure storage itself is unlikely to be the main source of the additional overhead.
> > >
> > > On large-scale graphs, although **dual_view_no_cl** does introduce extra computation and storage cost compared with **single-view**, it remains much lighter than the **full** configuration with the complete contrastive loss. More specifically, let the numbers of nodes, edges, feature dimension, and embedding dimension in the current client subgraph be $N$, $E$, $F$, and $d$, respectively. Then the main overheads of the three modes can be approximately understood as follows:
> > >
> > > - `single-view`: one subgraph input, one forward pass, and the storage of output embeddings, i.e., $M_{\text{single}} \approx O(NF + E + Nd)$, where $NF$ is the node feature input scale, $E$ is the sparse graph structure scale, and $Nd$ is the output embedding scale.
> > > - `dual_view_no_cl`: beyond `single-view`, this mode adds graph augmentation-based pseudo-labeling and consistency regularization. Its main extra cost comes from augmented node features, sparse edge indices, and augmented-view embedding storage, i.e., $M_{\text{dual}} \approx M_{\text{single}} + O(2NF + 2E + 2Nd)$.
> > > - `full`: based on `dual_view_no_cl`, this mode further adds the graph contrastive loss, whose main extra cost comes from pairwise similarity computation between node embeddings in the current client subgraph, i.e., $M_{\text{full}} \approx M_{\text{dual}} + O(N^2 d)$.
> > >
> > > From this view, the main extra overhead of `dual_view_no_cl` over `single-view` is still dominated by terms approximately linear in the current client subgraph size, while `full` introduces a higher-order term $O(N^2 d)$ due to the contrastive component.
> > >
> > > To further support the above analysis, we provide detailed overhead statistics in Tab. B at the anonymous link https://anonymous.4open.science/r/E-2E6B. Including  average and maximum forward training time, average and maximum peak GPU memory, average and maximum GPU memory of the graph contrastive-loss term, average memory during forward and backward propagation, and GPU memory occupied by augmented node features, edges, and embeddings(*Tab. A* explains the notation).  Note that some statistics are reported in **GB**, while others are reported in **MB**. The results show that the additional overhead of `dual_view_no_cl` is moderate, whereas the `full` configuration incurs a clear increase due to the graph contrastive-loss computation. Meanwhile, the GPU memory occupied by node features, edge structures, and stored embeddings remains relatively small. These observations indicate that, for large-scale graphs, `dual_view_no_cl` can retain the robustness benefits of dual-view enhancement while keeping the additional overhead manageable.
> > >
> > >
> > > **W2: CADF framework**
> > >
> > > **A2:** We thank you for this important follow-up. Due to space limitations, we respectfully refer you to our **Reply Rebuttal Comment** to Reviewer 9bLn, where the framework’s innovation, contributions, and improvements over prior work are detailed.
> > >
> > > **W3: Inductive evaluation**
> > >
> > > **A3:** We appreciate your guidance. Under the inductive setting, with 10 clients and noise level 0.3, we compared FedRGL with 12 methods on Flickr and Reddit. Detailed results are in Tab. C at https://anonymous.4open.science/r/E-2E6B
> > > . The trends match Tab. 2. FedRGL shows strong generalization and robustness across both datasets and noise types, thanks to CADF’s precise noisy-node filtering and pseudo-label supervision.

---

### Official Review · Reviewer_9bLn · 2026-03-13

**Soundness:** 3
**Presentation:** 3
**Significance:** 3
**Originality:** 2
**Overall Recommendation:** 4
**Confidence:** 3

**Summary:**

The paper studies robust federated graph learning under label noise, where incorrect labels within client-side subgraphs can degrade the quality of the aggregated global model. This problem is challenging because each client only observes a partial subgraph and the noise distribution may vary across clients. To address this issue, the authors propose FedRGL, a framework that attempts to detect noisy nodes by jointly leveraging global model predictions and local structural information. To further mitigate the impact of noisy supervision during training, the method incorporates contrastive learning to refine node representations and introduces a client-level reliability signal during aggregation to account for differences in client data quality. Experiments on several graph benchmarks are conducted to evaluate the robustness of the proposed approach under different levels of label noise.

**Compliance With Llm Reviewing Policy:**

Affirmed.

**Key Questions For Authors:**

1. Could the authors clarify what fundamentally distinguishes the proposed approach from existing noisy-label learning or robust federated graph learning methods, beyond the integration of these components?

2. Could the authors provide a more direct evaluation of the noise detection performance (e.g., precision/recall or other identification metrics) to verify whether the filtering module reliably detects corrupted labels under different noise settings?

3. How does the method avoid reinforcing bias when the global model itself may already be affected by noisy labels during early training stages? Additional analysis of this potential feedback effect would help clarify the robustness of the filtering strategy.

**Limitations:**

Yes.

**Strengths And Weaknesses:**

**Strengths**:
1. The paper addresses the problem of label noise in federated graph learning, which is a relevant challenge for real-world decentralized graph data scenarios where noisy supervision may propagate across clients.
2. The proposed framework attempts to combine global prediction signals and local structural information to identify noisy nodes during federated training.
3. The method integrates noise filtering with representation learning components, including contrastive learning and pseudo-label refinement, aiming to mitigate the impact of corrupted supervision during local training.
4. The empirical evaluation considers multiple benchmark graph datasets and different noise settings, providing evidence on the robustness of the proposed approach in federated graph learning scenarios.


**Weaknesses**:
1. The overall framework integrates several existing components such as dual-view noise filtering, contrastive learning, pseudo-label supervision, and entropy-based aggregation. While the integration is reasonable, the paper does not clearly demonstrate a fundamentally new algorithmic principle beyond combining these techniques for the federated graph learning setting.
2. The method heavily relies on the proposed noisy node filtering mechanism. However, the empirical evaluation mainly focuses on downstream classification accuracy and does not directly assess the quality of noise detection. Providing quantitative metrics (e.g., precision/recall of noisy node identification) would help verify whether the filtering mechanism reliably distinguishes corrupted labels.
3. The filtering mechanism relies partly on predictions from the global model to identify noisy nodes. However, when the global model itself is influenced by noisy labels during early training stages, this feedback loop may introduce bias in the filtering process. The paper does not analyze this potential issue.
4. The experimental evaluation relies on synthetic label noise (uniform, pair, and instance-dependent noise) generated from clean benchmark datasets. Although this protocol is common in noisy label studies, evaluating the approach on datasets with naturally occurring annotation errors would provide stronger evidence of its practical robustness.

---

> ### Author Rebuttal · Authors · 2026-03-31
>
> **Dear Reviewer 9bLn,**
>
> Thank you for your review and comments. We appreciate your recognition of the problem relevance, the experimental evaluation, and the robustness of our method. Below we address your main concerns point by point.
>
> **W1&Q1: Novelty and differences from prior methods**
>
> **A1:** Thank you for this question. FedRGL is not a simple combination of existing techniques, but is designed for noisy subgraph FL, where noisy labels, missing global structure, and class imbalance coexist. As shown in Fig. 1, each client only holds a local subgraph with imbalanced classes, and existing FGL methods degrade under label noise. Fig. 2 further shows that neither vision-oriented FNL methods nor centralized GNL methods can be directly adapted well to this setting. More importantly, **CADF is the core contribution**. Unlike methods based on unified thresholds, single-view filtering, or complete graph structures, CADF jointly uses the global model view and the local structural view with class-aware dynamic thresholds, making it better suited to federated subgraphs with inconsistent class learning rates and incomplete structures. Its difference from related methods is clarified in p.3, right column, lines 151–152 and Appendix B, and its advantage is further verified in Appendix Fig. 8(c).
>
> After noisy-node detection, FedRGL further introduces high-confidence pseudo-label supervision and predictive-entropy-based aggregation reweighting to address supervision recovery and robust aggregation, so they are not loose add-ons. In addition, Tab. 3 verifies the plug-and-play ability of CADF on FGSSL, FedTAD, and FedSPA, while Appendix O further extends it to FedSage+, FedLoG, FedGTA, and FedATH, covering 7 subgraph FL frameworks in total. Therefore, FedRGL achieves stronger robustness under noisy subgraph FL and shows good generalization potential.
>
> **W2&Q2: Direct evidence of CADF filtering**
>
> **A2:** As reported in Table 1 on p.5 of the main paper, the noisy-node filtering ability of CADF has already been directly quantified through the noise node filtering success rate on Cora with 0.5 uniform noise across different rounds and clients. The results show that CADF becomes progressively more accurate during training. To further quantify the capability of CADF, we additionally report its filtering accuracy on CS at round 50 under 0.1/0.3/0.5 noise across different clients (see Tab. A in the anonymous link: https://anonymous.4open.science/r/Tab-B-4708/README.md). These results further show stable performance across clients and noise rates.
>
> **W3&Q3: Early-stage global robustness**
>
> **A3:** Thank you for this question. This can be understood by the memorization effect in noisy-label learning, where models fit reliable patterns first and noisy labels later [1]. Fig. 2(a) is consistent with this: under noisy subgraph FL, existing FNL baselines usually gain accuracy quickly at the beginning, but as training proceeds, clean and noisy nodes become increasingly entangled, making robust generalization harder to maintain, some methods even show clear late-stage degradation.
>
> To verify this, we visualize the average clean/noisy-node losses on Cora with 0.5 noise for FedRGL and FedAvg (see Fig. A in https://anonymous.4open.science/r/Tab-B-4708/README.md). In both methods, clean-node loss drops faster early on, showing that the model first fits relatively reliable clean nodes. As training continues, FedAvg keeps reducing the noisy-node loss, indicating increasing memorization and weaker clean/noisy separability. **In contrast**, FedRGL keeps decreasing the clean-node loss, while the noisy-node loss rises later and remains clearly separated. This suggests that FedRGL preserves clean/noisy separability during training and thus reduces the disturbance of noisy nodes to the global model. As shown in App. G, FedRGL does not activate filtering, pseudo-label recovery, or quality-aware aggregation from the first round. It first uses a warm-up stage, and enables these robust operations only after the global model becomes initially discriminative.
>
> [1] Learning From Noisy Labels With Deep Neural Networks: A Survey, TNNLS 2023.
>
> **W4: Natural noisy graph data**
>
> **A4:** Evaluation on natural noisy graph data would further strengthen the paper. However, according to our investigation, publicly available and widely adopted natural noisy graph-structured datasets are **still lacking**, so existing noisy-label learning studies on graph data mainly rely on synthetic corruption. Our experiments follow this common practice by considering three representative noise types: uniform, pair, and instance-dependent noise. Although natural noisy graph benchmarks are not included, we strengthen the evaluation from multiple perspectives, including small-/large-scale datasets, multiple noise types, and additional analysis under different client numbers, noisy-client ratios, and participation rates. We will clarify this boundary more explicitly in the revision.

---

> > ### Author Rebuttal · Reviewer_9bLn · 2026-04-03
> >
> > Thank you to the authors for the detailed response to the concerns I raised. I appreciate the clarifications provided, particularly regarding the evaluation on noisy graph data and the effectiveness of the CADF filtering mechanism, which have been addressed more clearly.
> >
> > However, as stated by the authors, the core contribution of this work lies in the proposed CADF module that combines the global model view and the local structural view. In my assessment, this design appears to be largely a refinement and integration of existing techniques rather than a fundamentally new algorithmic principle. While the combination is reasonable and empirically effective, the level of conceptual novelty remains limited.
> >
> > I acknowledge that the experimental evaluation is comprehensive and well-executed. Nevertheless, given the **limited novelty**, I believe my current score appropriately reflects the overall contribution of the paper.

---

> > > ### Author Response · Authors · 2026-04-06
> > >
> > > **Dear Reviewer 9bLn,**
> > >
> > > Thank you for your reply and questions. Below, we explain our method’s novelty and contributions in three aspects.
> > >
> > > **1. Research motivation.**
> > >
> > > FedRGL targets robust subgraph-FL under the coexistence of noisy labels, class imbalance, and topological heterogeneity across clients, a setting still lacking dedicated robust designs. We acknowledge that the local label propagation, predictive entropy, and graph augmentation involved in FedRGL build on existing methods. However, FedRGL further introduces targeted refinements to better address the multiple challenges in noisy subgraph-FL. Specifically, in the Introduction and **Fig. 1** on p. 2, we show the difficulty of improving global robustness when local client subgraphs are simultaneously affected by noisy labels, class imbalance, and topological heterogeneity across clients, and empirically show the degradation of several existing federated graph learning (FGL) methods in this setting. In **Fig. 2** on p. 2, we further evaluate representative image-domain federated noisy-label learning (FNL) methods and centralized graph noisy-label learning (GNL) methods in the subgraph-FL setting. The results show that existing FNL and GNL methods cannot be effectively transferred. Based on these observations, FedRGL further introduces class-aware dynamic thresholds and dual-view consistency for more precise noisy-node identification under heterogeneous and imbalanced subgraphs, and further incorporates high-confidence pseudo-labeling and robust aggregation.
> > >
> > > **2. Connections to and differences from prior work.**
> > >
> > > The core module is **CADF**, which uses the global-model prediction loss and the propagated loss from local subgraph label propagation, under *class-aware dynamic thresholds*, to locate and filter noisy nodes through dual-view consistency.
> > >
> > > - Like most FNL methods, FedRGL also uses loss information for noisy-label filtering. The difference is that existing methods usually adopt a *GMM-based strategy* (FedCorr) or *fixed thresholds* (FedNoRo), which treat different classes in a relatively unified manner. In contrast, in noisy subgraph-FL, class imbalance and structural incompleteness cause different classes to receive different neighborhood information and exhibit different learning rates, so unified thresholds often fail to distinguish *hard-but-clean nodes* from truly noisy nodes. We therefore propose **class-aware dynamic thresholds** to more precisely identify noisy nodes under heterogeneous and imbalanced subgraphs. This motivation is described in **Sec. 3.2.1** on p. 4.
> > >
> > > - The local subgraph view in CADF is built on basic label propagation, and **we explicitly acknowledge this connection**. We provide a detailed comparison in *App. B* . The main differences are: 1) CADF does not require clean labeled nodes as prior knowledge, whereas **GNN-Cleaner** and **R2LP** do, although such clean supervision is difficult to obtain in real scenarios; 2) before local label propagation, CADF uses the robust global model to evaluate and correct the initial node labels, whereas **ERASE** directly uses the initial labels; 3) after propagation, prior methods usually rely on *one-hot label information*, while CADF uses propagated *soft labels* with original labels to compute the local loss, thus incorporating structural information into filtering.
> > >
> > > After filtering noisy nodes, FedRGL adopts graph augmentation from existing graph contrastive learning methods to assign pseudo-labels to some high confidence noisy nodes, and uses a contrastive objective to improve robust representation learning. This design is explained in **Sec. 3.2.2** on p. 5, where the method description also explicitly cites **GRACE**. In addition, since most existing subgraph-FL methods are built under the *transductive setting*, we design a local training-quality metric based on the **average predictive entropy of unlabeled nodes** for robust server-side aggregation. The motivation and design considerations are discussed in **Sec. 3.2.3** on p. 6.
> > >
> > > **3. Theory and evidence.**
> > >
> > > On p. 5, Proposition 3.1 provides an upper-bound analysis of CADF’s noisy-node filtering error. It shows that class-aware dynamic thresholds can constrain the probability that noisy nodes are mistakenly selected as clean, while the intersection under dual-view consistency can further reduce it. *Tab. 1* reports CADF’s noisy-node filtering success rates across communication rounds and clients. **Fig. 8(c)** in the appendix shows that using dynamic thresholds in only one view, or degenerating to static GMM-style filtering, is clearly weaker than full CADF. As a plug-and-play module, CADF substantially improves the noise robustness of **seven existing federated subgraph learning frameworks**. We also provide **anonymous open-source code** and each dataset’s hyperparameter settings to support reproducibility.
> > >
> > > We hope the above explanation helps clarify the novelty and contribution of our method. Thank you again.

---

### Official Review · Reviewer_9mgj · 2026-03-13

**Soundness:** 2
**Presentation:** 2
**Significance:** 2
**Originality:** 2
**Overall Recommendation:** 3
**Confidence:** 5

**Summary:**

This paper studies the label noise problem in federated subgraph learning (subgraph FL), proposing the FedRGL framework, which includes three core modules: (1) CADF: dual-view consistency noise node filtering based on class-aware dynamic thresholding (global model view + local structure view); (2) assigning high-confidence pseudo labels to noisy nodes using graph contrastive learning; (3) server-side aggregation reweighting based on the prediction entropy of unlabelled nodes. Extensive experiments are conducted on multiple graph datasets.

**Compliance With Llm Reviewing Policy:**

Affirmed.

**Final Justification:**

I have discussed the rebuttal quality in the acknowledgment. Overall, the paper solves an interesting problem, but it has fewer theoretical foundations and more decorative components for the robustness design. If it has convergence analysis and explainable design for the proposed components that are more unified, it is worth acceptance. For now, I recommend weak rejection.

**Key Questions For Authors:**

1. Does line 166 means the noise rate is assumed to be known before training? Does FedRGL rely on prior knowledge of the noise rate?
2. What is the influence of label propagation step k for the performance?

**Limitations:**

Yes.

**Strengths And Weaknesses:**

Strengths:
1. The problem motivation is well-founded and significant. The paper effectively demonstrates the shortcomings of existing FGL and FNL methods in noisy graph data scenarios through Figures 1 and 2, providing clear motivation.
2. The experiments cover a wide range. They include 8 datasets, 3 types of noise, different noise rates and number of clients, with baseline methods covering 12 approaches across three categories, making the comparison relatively comprehensive. Meanwhile, FedRGL leads with a significant advantage in most settings.
3. The plug-and-play design of the CADF module is commendable.
4. The paper is well written and features good visual illustrations.

Weaknesses:
1. The global model aggregation is reweighted by predictive entropy. While this seems to enhance robustness under noisy label conditions, the traditional data size is directly ignored. Based on conventional FL, local data sizes play an important role in balancing aggregation weights. The reviewer is uncertain about completely replacing this factor. Further discussion on this design choice is expected.
2. The graph contrastive learning component almost completely follows GRACE. Its uniqueness under the noisy label condition is not highlighted. The ablation of this component indicates its minor help on the performance.
3. The proposed FedRGL introduces additional communication overhead during each training round. A thorough discussion on this is necessary, and a numerical runtime comparison with analysis is also suggested.
4. What is the actual quality of the pseudo relabeling? Table 1 reports only the noise filtering success rate, not the pseudo-label quality.

---

> ### Author Rebuttal · Authors · 2026-03-31
>
> **Dear Reviewer 9mgj,**
>
> We thank you for your constructive comments and positive assessment of the problem, experiments, and CADF design. Below we respond to each concern.
>
> **W1: Entropy-based aggregation.**
>
> **A1:** Thank you for the question. FedRGL adopts a two-stage aggregation strategy: size-based aggregation during warm-up for stable early optimization, and entropy-based aggregation afterward. This is described in Appendix G. It is also explained in the main text (p. 6, right column, lines. 302--308), where we state that the warm-up phase does not use predictive-entropy-based server reweighting. The corresponding implementation can be directly checked in `train_FedRGL_Cora.py`, Lines 263--282.
> Under Louvain partition, client sizes are relatively similar (**Appendix B, Fig. 7(a)**), so data size offers limited discrimination in later rounds. By contrast, structural heterogeneity, class imbalance, and noisy labels dominate local update quality. Therefore, average predictive entropy better reflects local prediction uncertainty and update reliability.
>
> To further examine this issue, under the same settings as Tab. 2, we tested a hybrid variant, **FedRGL#**, which combines client size with entropy-based weighting. To save rebuttal space, we report three representative datasets below and provide the full results on all six datasets in the anonymous repository: https://anonymous.4open.science/r/Tab-A-FB43/README.md. As shown in Tab. A, **FedRGL#** may bring slight gains in the clean setting, but generally degrades performance under noisy settings, especially under Pair noise. This suggests that directly incorporating client size into entropy-based weighting may weaken the quality-aware suppression of low-quality updates.
>
> *Tab. A. Aggregation comparison (N/U/P: Normal/Uniform/Pair)*
>
> |Data|Cora|||CiteSeer|||Physics|||
> |-|-|-|-|-|-|-|-|-|-|
> |Meth|N|U|P|N|U|P|N|U|P|
> |FedRGL|82.12|78.75|75.14|71.52|66.08|63.22|93.62|89.66|86.39|
> |FedRGL#|82.03|78.03|74.21|71.61|65.33|61.82|93.85|88.51|84.95|
>
> **W2: Contrastive learning module.**
>
> **A2:** We would like to clarify that the core contribution of this work is **CADF**, a plug-and-play noisy-node filtering module for local client subgraphs. By contrast, the graph contrastive module is auxiliary rather than the main novelty. It uses graph augmentation to produce dual-view predictions and generate high-confidence pseudo-labels for nodes filtered by CADF, while the contrastive loss further enforces cross-view consistency (Sec. 3.2.2, p. 5; GRACE is cited in lines 274--275). Importantly, **w/o CL** in Fig. 6(b) keeps graph augmentation and pseudo-label generation, but removes only the contrastive loss. Under this setting, performance drops by **0.60%** and **0.79%** under the two noise types, respectively, showing that CL provides additional gains, but is not the core mechanism for noisy-node filtering.
>
> **W3: Communication and runtime overhead.**
>
> **A3:** Thank you for raising this point. We have already provided theoretical analysis and numerical validation in Appendix. H.1. In communication, FedRGL uploads only one additional scalar per client per round, i.e., the average predictive entropy. Therefore, its per-client communication complexity is $O(|\theta|+1)$, which is asymptotically equivalent to FedAvg’s $O(|\theta|)$. In runtime, the extra cost mainly comes from CADF. Tab. 5 reports the total training time over 100 communication rounds, while Tab. 2 shows that this cost brings stronger robustness under noisy labels. Moreover, Tab. 6 shows that applying CADF once every 10 rounds significantly reduces training time with only slight accuracy loss. Please refer to App. H.1 for further details.
>
> **W4: Pseudo-label quality.**
>
> **A4:** Thank you for the question. To validate pseudo-label quality, we conducted an additional experiment on **CS** with **10 clients** under **Uniform 0.5** noise. Due to limited rebuttal space, the detailed results are provided as **Tab. B** at <https://anonymous.4open.science/r/Tab-A-FB43/README.md>. The results show that the average accuracy of high-confidence pseudo-labels rises from **82.95%**, **87.08%**, to **92.62%** at rounds **30**, **50**, and **70**, respectively, indicating that the pseudo-labels are reliable and improve as training proceeds.
>
> **Q1: Need a known noise rate?**
>
> **A1:** FedRGL does **not** require the true noise rate to be known during training. We clarify that Line 166 refers only to the experimental noise-rate sampling strategy for constructing heterogeneous client noise, rather than a prerequisite of the method itself. For reproducibility, Appendix. J further provides the corresponding noise types and sampling ranges used in our experiments.
>
> **Q2: Sensitivity to $k$.**
>
> **A2:** The sensitivity analysis of $k$ is provided in Appendix. F, Fig. 10(a)(b). The results show that FedRGL remains generally stable w.r.t. $k$, with only slight fluctuations on a few datasets, indicating that the adopted setting is robust.

---

> > ### Author Rebuttal · Reviewer_9mgj · 2026-04-04
> >
> > Thanks for the author's rebuttal. Most of my concerns are resolved.
> > I understand the experimental results as provided in the rebuttal may indicate that the data size weights may cause worse performance. But why is it theoretically? Cause the data size is the key between global loss and local loss initially, and it is independent of the label noise.
> > Meanwhile,  the discussion of W2 is not convincing to me. If the gain is limited, why use GRACE?
> > Furthermore, I double-check the paper, finding that the convergence analysis is missing from the algorithm. How can the method be guaranteed for convergence under the new aggregation design as well as the label correction? After reading other reviewers' comments, I may reconsider my rate for relatively truly reflecting the contribution of the paper.

---

> > > ### Author Response · Authors · 2026-04-06
> > >
> > > **Dear Reviewer 9mgj,**
> > >
> > > Thank you for your follow-up response. Our detailed replies are below.
> > >
> > > **W1: Aggregation.**
> > >
> > > **A1:** We agree that client size is important in conventional federated learning, where it is often a reasonable aggregation factor when client updates do not differ much in reliability. However, our setting is noisy subgraph FL rather than clean-label FL. Under label noise, class imbalance, and topological heterogeneity across clients, client size cannot always reflect update reliability well. Therefore, Eq. (13) uses the average predictive entropy of unlabeled nodes for aggregation, since it better measures local model quality. In our previous response, we constructed a hybrid variant, FedRGL#, as $W^{t+1}=\sum_{m=1}^{M}\frac{V_m H_m}{\sum_{m=1}^{M}V_m H_m}w_m^t,$
> > > where $V_m$ is client size and $H_m$ is the reciprocal of average predictive entropy. Although label corruption does not change client size, reintroducing $V_m$ after warm-up still changes the aggregation weights at the server side, weakens the role of predictive entropy as a quality signal, and may amplify the influence of large but low-quality clients. Even under the Louvain partition, where client-size differences are not large, reintroducing client size can still affect the aggregation weights in each round.This may prevent the global model from obtaining a more favorable aggregation result, affect the next communication round of training, and eventually cause some drop in accuracy. This conclusion is also supported by the supplementary results in our previous response.
> > >
> > > In addition，We adopt a two-stage aggregation strategy because noisy-label learning typically exhibits a memorization effect: models first fit cleaner patterns and then gradually fit noisy labels. The same phenomenon also appears in noisy subgraph FL, as further discussed in our response to Q3 in Reviewer 9bLn’s first-round review. Therefore, we use client-size-based aggregation for warm-up and switch to predictive-entropy-based aggregation later, when update reliability differences become clearer. This is also reflected in the anonymous code (`train_FedRGL_Cora.py`, lines 263-282).
> > >
> > > **W2: GRACE Method.**
> > >
> > > **A2:** We apologize for the earlier confusion. The introduction of **GRACE** in this work has two purposes:
> > > 1) graph augmentation constructs dual-view predictions for high-confidence pseudo-label enhancement on suspicious noisy nodes identified by CADF;
> > > 2) the extra contrastive-loss term further regularizes encoder training.
> > >
> > > Therefore, w/o CL in Fig. 6(c) indicates that graph augmentation is retained and only the extra contrastive-loss(*CL*) term is removed. This ablation tests the marginal gain of the CL term itself, rather than the effectiveness of the whole GRACE-based dual-view design. We additionally provide results on Cora after removing graph augmentation, as shown in **Fig. A** at https://anonymous.4open.science/r/F-8DB3. Under the two noisy settings, removing graph augmentation causes FedRGL to drop by **2.87%** and **3.29%**, respectively. This shows that although the extra CL term brings only limited improvement, the graph-augmentation-based dual-view design remains important. We will add this additional ablation in the revised version to make this distinction clearer.
> > >
> > > **W3 Convergence.**
> > >
> > > **A3:** Thank you for your guidance. Due to space limits, we give only the main idea here and will add the full proof in the revised version. On **page 5**, we have already presented and proved **Proposition 3.1**, which gives a theoretical bound on **CADF** filtering error. In particular, **Eq. (14)** bounds the probability that noisy nodes are falsely retained in a single view, while **Eq. (16)** shows that global/local dual-consistency further shrinks this probability. Based on this proposition, the error introduced by subsequent high-confidence pseudo-label (**HCPL**) can be modeled as a bounded local-gradient bias.
> > >
> > > Specifically, let $\tilde g_m^t$ denote the local gradient with HCPL and $g_m^t$ the ideal one, with $\|\tilde g_m^t-g_m^t\|\le \beta_t$, where $\beta_t$ captures the bias from CADF filtering error and subsequent pseudo-label error. Meanwhile, **Eq. (13)** gives the predictive-entropy aggregation rule used in training. This mechanism affects how local updates are accumulated at the server side and helps suppress low-quality client updates. Under standard assumptions including $L$-smoothness, bounded stochastic-gradient variance, and bounded client heterogeneity, the overall optimization admits the corresponding convergence bound, where the effects related to HCPL, JS consistency, and the extra CL term can all be absorbed into this additional bounded bias term. Therefore, when this bias remains sufficiently small, FedRGL still converges to a neighborhood of a stationary point. In addition, **Fig. B** at https://anonymous.4open.science/r/F-8DB3 provides the test-accuracy convergence curves, which further verify the convergence of FedRGL.

---

### Decision · Program_Chairs · 2026-04-30

**Decision:**

Accept (regular)

**Comment:**

This paper introduces FedRGL, a robust federated graph learning framework that addresses label noise in federated subgraph learning. The proposed Class-Aware Dual-Consistency Filtering (CADF) module provides a robust selection mechanism for filtering noise and can be seamlessly integrated in a plug-and-play manner into various existing frameworks. In their reviews, the reviewers unanimously agreed that the problem addressed in this paper is clearly motivated and of critical importance. Furthermore, extensive and rigorous experiments conducted on real-world graph datasets demonstrate the method's strong robustness across various noise types and different client scales. In their rebuttal, the authors provided detailed and effective responses to initial concerns regarding computational overhead for large-scale graph data and the method's applicability within the specific experimental settings, thereby further strengthening the paper's empirical contributions. Although some questions remain regarding the novelty of the individual components, given its evident practical value and strong experimental validation, I recommend this paper for acceptance as a poster presentation.